



# Impact of horizontal resolution on global ocean-sea-ice model simulations based on the experimental protocols of the Ocean Model Intercomparison Project phase 2 (OMIP-2)

Eric P. Chassignet[1], Stephen G. Yeager[2], Baylor Fox-Kemper[3], Alexandra Bozec[1], Frederic Castruccio[2],
Gokhan Danabasoglu[2], Who M. Kim[2], Nikolay Koldunov[4], Yiwen Li[5], Pengfei Lin[5], Hailong Liu[5],
Dmitry Sein[4,6], Dmitry Sidorenko[4], Qiang Wang[4], and Xiaobiao Xu[1]

[1]Center for Ocean-Atmospheric Prediction Studies, Florida State University, Tallahassee, FL, USA
[2]National Center for Atmospheric Research, Boulder, CO, USA
[3]Brown University, New Providence, RI, USA
[4]Alfred-Wegener-Institut Helmholtz-Zentrum für Polar- und Meeresforschung (AWI), Bremerhaven, Germany
[5]State Key Laboratory of Numerical Modeling for Atmospheric Sciences and Geophysical Fluid Dynamics, Institute of
Atmospheric Physics, Chinese Academy of Sciences, Beijing, China
[6]Shirshov Institute of Oceanology, Russian Academy of Science, Moscow, Russia

*Correspondence to*: Eric P. Chassignet (echassignet@fsu.edu)

**Abstract.** This paper presents global comparisons of fundamental global climate variables from a suite of four pairs of matched low- and high-resolution ocean and sea-ice simulations that are obtained following the OMIP-2 protocol (Griffies et al., 2016) and integrated for one cycle (1958-2018) of the JRA55-do atmospheric state and runoff dataset (Tsujino et al., 2018). Our goal is to assess the robustness of climate-relevant improvements in ocean simulations (mean and variability) associated with moving from coarse (~1º) to eddy-resolving (~0.1º) horizontal resolutions. The models are diverse in their numerics and parameterizations, but each low-resolution and high-resolution pair of models is matched so as to isolate, to the extent possible, the effects of horizontal resolution. A variety of observational datasets are used to assess the fidelity of simulated temperature and salinity, sea surface height, kinetic energy, heat and volume transports, and sea ice distribution. This paper provides a crucial benchmark for future studies comparing and improving different schemes in any of the models used in this study or similar ones. The biases in the low-resolution simulations are familiar and their gross features – position, strength, and variability of western boundary currents, equatorial currents, and Antarctic Circumpolar Current – are significantly improved in the high-resolution models. However, despite the fact that the high-resolution models "resolve'' most of these features, the improvements in temperature or salinity are inconsistent among the different model families and some regions show increased bias over their low-resolution counterparts. Greatly enhanced horizontal resolution does not deliver unambiguous bias improvement in all regions for all models.

## 1 Introduction

A key decision in climate model design is the spatial resolution of different components. The global scope of the integrations and the centennial to millennial timescales and multiple scenarios required to capture changes in the climate set the problem; the spatial resolution is therefore the result of available computing. This decade, computing has become



sufficiently powerful to make mesoscale-rich resolution affordable in the ocean and sea-ice models over most of the earth, which allows the ocean model to simulate more intense internal variability than a lower resolution model. As this new regime of coupled modeling is entered, it is important to understand both the behavior of ocean and sea ice models in a controlled framework, and the benefits and challenges that come with higher resolution. This work introduces a set of matched numerical simulations in low- and high-resolution ocean and sea-ice models with the latest forcing protocol. It is anticipated that these

results will inform fully coupled modeling studies where ocean resolution varies, and also that follow-on studies will build on our results to examine process and regional detail.

In 2016, an international group of ocean modelers behind the development and analysis of global ocean−sea-ice models used as a component of the earth system models in CMIP6 proposed an Ocean Model Intercomparison Project (OMIP) (Griffies et al., 2016). The essential element behind the OMIP is a common set of atmospheric and river runoff datasets for computing

surface boundary fluxes to drive the ocean–sea-ice models, many of which are used as components of coupled climate system models. The OMIP protocol is an outcome of the Coordinated Ocean–sea-ice Reference Experiments (CORE) which assessed the performance of ocean−sea-ice models (Griffies et al., 2009; Danabasoglu et al., 2014, Griffies et al., 2014; Downes et al., 2015; Farneti et al., 2015; Danabasoglu et al., 2016; Wang et al., 2016a, 2016b; Ilicak et al., 2016; Tseng et al., 2016; Rahaman et al., 2019) using the atmospheric and river runoff dataset of Large and Yeager (2009).  However, this dataset has not been

updated since 2009 and a new dataset (JRA55-do; Tsujino et al., 2018) has been developed for the OMIP based on the Japanese Reanalysis (JRA-55) product from Kobayashi et al. (2015) to ensure that it is regularly updated. This raw reanalysis product has been substantially adjusted to match reference states based on observations or ensemble mean of other atmospheric reanalysis products as detailed in Tsujino et al. (2018) to create a suitable forcing dataset for ocean and sea-ice models, referred to as JRA55-do. The continental river discharge is provided by a river-routing model forced by input runoff from the land-

surface component of JRA-55 adjusted to ensure similar long-term variabilities as in the CORE dataset (Suzuki et al., 2018). Runoff of ice-sheet and glaciers from Greenland (Bamber et al., 2012; Bamber et al., 2018) and Antarctica (Depoorter et al., 2013) are also incorporated. Tsujino et al. (2020) presents an evaluation of the simulations from CMIP6-class global ocean– sea-ice models forced with the JRA55-do datasets. This effort compares CORE-forced (i.e., OMIP-1) and JRA55-do-forced (i.e., OMIP-2) simulations considering metrics commonly used in the evaluation of global ocean–sea-ice models to assess

model biases.

Many features are very similar between OMIP-1 and OMIP-2 simulations, but Tsujino et al. (2020) identify many improvements in the simulated fields in transitioning from OMIP-1 to OMIP-2. For example, the sea surface temperature of the OMIP-2 simulations reproduce the global warming hiatus in the 2000s and the recent observed warming, which are absent in OMIP-1 partly because the OMIP-1 forcing stopped in 2009. The low bias in the sea-ice area fraction in summer of both

hemispheres in OMIP-1 is significantly improved in OMIP-2. The overall reproducibility of both seasonal and interannual variation in sea surface temperature and sea surface height (dynamic sea level) is also improved in OMIP-2. Tsujino et al. (2020) attributes many of the remaining model biases either to errors in representing important processes in ocean–sea-ice models, some of which are expected to be mitigated by taking finer horizontal and/or vertical resolutions, or to shared biases



in the atmospheric forcing. In this paper, we make a first attempt at quantifying the impacts of the models' horizontal resolution
on biases.

Our goal is to assess the robustness of climate-relevant improvements in ocean simulations (mean and variability)
associated with moving from coarse (~1º) to eddy-resolving (~0.1º) horizontal resolutions. Using the same atmospheric forcing
(JRA55-do) for both low- and high-resolution configurations, we perform a multi-model analysis to identify the robust
differences and improvements associated with increased resolution given the same forcing datasets. Within the ocean modeling
community, it is usually assumed that high-resolution simulations should in general produce better results than low-resolution
ones (Fox-Kemper et al., 2019). While this is clearly the case for surface currents and internal variability, we will show that
greatly enhanced horizontal resolution does not necessarily deliver unambiguous bias improvement in temperature and salinity
in all regions. It is important to note several caveats when interpreting the results presented in this paper: First, this is based on
a limited number of numerical models (four) and, second, because of the large computational cost associated with the high-
resolution runs (factor 1000 more expensive), only one JRA55-do cycle (1958-2018) is analyzed in this paper (versus 6 cycles
for the coarse-resolution runs of Tsujino et al. (2020)). Also, because of the short integration time, some of the results may not
be robust (AMOC variability, deep ocean circulation, etc.) (Danabasoglu et al., 2016). The layout of the paper is as follows.
Section 3 highlight differences in the magnitude of the models' drift while Section 4 focuses on the detrended interannual to
decadal variability and the differences in the modeled ocean climates. The results are summarized and discussed in the final
section.

## 2 Model's description

Because the goal is to identify the robust differences and improvements associated with increased horizontal resolution
given the same forcing datasets, most of the participating modeling groups configured their high-resolution configuration with
similar parameters to that of the coarse-resolution configuration (Table 1). It is important to note that not all the models used
the same climatology for the initial conditions, nor did they use the same wind stress formulation (absolute versus relative
winds). When evaluating the drift of a numerical simulation, it is performed with respect to the climatology used to initialize
the run.

### 2.1 FSU-HYCOM

The FSU-HYCOM is a global configuration of the HYbrid Coordinate Ocean Model (HYCOM) (Bleck, 2002;
Chassignet et al., 2003; Halliwell, 2004). For the coarse resolution configuration, the grid is a tripolar Arakawa C-grid of 0.72º
horizontal resolution with refinement to 0.33º at the equator (500 cells in the zonal direction and 382 in the meridional
direction). The 2-minute NAVO/Naval Research Laboratory DBDB2 dataset provides the bottom topography. Forty-one
hybrid coordinate layers whose $\sigma_2$ target densities range from 17.00 to 37.42 kg/m$^3$ are used. The vertical discretization
combines fixed pressure coordinates in the mixed layer and unstratified regions, isopycnic coordinates in the stratified open
ocean, and terrain-following coordinates over shallow coastal regions. The initial conditions in (potential) temperature and



salinity are given by the Generalized Digital Environmental Model (GDEM4, Teague et al., 1990; Carnes, 2009)). The sea-ice component is CICE (Hunke and Lipscomb, 2010). The Large and Yeager (2004) bulk formulation is used for turbulent air-sea fluxes except for the surface wind-stress that is calculated without surface currents (absolute wind stress). No restoration is applied on the sea surface temperature. The surface salinity is restored to the monthly GDEM4 climatology over the entire

domain with a salinity piston velocity of 50 m/60 days. The salinity flux is globally normalized at each time step. Vertical mixing is the KPP model (Large et al., 1994) with a background diffusion of $3\ 10^{-5}$ m$^2$/s. The horizontal advection uses a second-order flux corrected transport scheme. A Laplacian diffusion of $0.03\Delta x$ is applied on temperature and salinity and a combination of Laplacian ($0.03\Delta x$) and biharmonic ($0.05\Delta x^3$) dissipation is applied on the velocities (see Table 1). The model baroclinic and barotropic time steps are 1800s (leap-frog) and 56.25s (explicit) respectively. Interface height smoothing by a

biharmonic operator is used to correspond to Gent and McWilliams (1990), with a mixing coefficient determined by the grid spacing $\Delta x$ (in m) times a velocity scale of 0.02 m s$^{-1}$ everywhere, except in the North Pacific and North Atlantic where a Laplacian operator with a velocity scale of 0.01 m s$^{-1}$ is used. Gent and McWilliams (1990) is not implemented where the FSU-HYCOM has coordinate surfaces aligned with constant pressure (mostly in the upper ocean mixed layer) and there is no rotation of the lateral diffusion along neutral surfaces. There is no parameterization of the overflows.

For the global high-resolution configuration, the grid is a tripolar Arakawa C-grid of 0.08º (1/12º) horizontal resolution (4500 cells in the zonal direction and 3298 in meridional direction). The model bathymetry is derived from the 30 arc-second GEBCO08 dataset. A vertical resolution of 36 hybrid layers, with $\sigma_2$ target densities ranging from 26.00 to 37.24 kg/m$^3$, is used. The initial conditions in temperature and salinity are given by GDEM4. Similar to the low-resolution configuration, the turbulent air-sea fluxes are computed using the Large and Yeager (2004) bulk formulation and surface wind-stress calculation

does not include the surface currents. No restoration is applied on the sea surface temperature. The surface salinity is restored to the monthly GDEM4 climatology over the entire domain with a salinity piston velocity of 50 m/60 days, and the salinity flux at each time step is adjusted to ensure a net global flux of zero. The KPP model (Large et al., 1994) with a background diffusivity of $10^{-5}$ m$^2$/s provides vertical mixing. The horizontal advection uses a second-order flux corrected transport scheme. The horizontal diffusion uses a combination of constant Laplacian diffusion of 20 m$^2$/s and a biharmonic dissipation of $0.01\Delta x^3$

for velocities; and the same biharmonic diffusion for tracers. The model baroclinic and barotropic time steps are 150s (leap-frog) and 5 s (explicit), respectively. An interface height smoothing is applied through a biharmonic operator (with a velocity scale of 0.015 m s$^{-1}$). There is no parameterization of the overflows.

## 2.2 NCAR-POP

The NCAR contribution is based on the Community Earth System Model version 2 (CESM2; Danabasoglu et al. 2020).

The ocean component is based on the Parallel Ocean Program version 2 (POP2; Smith et al. 2010), but features several modifications to model physics and numerics including improved treatment of continental freshwater discharge into unresolved estuaries (Sun et al., 2019) and a new parameterization of Langmuir mixing (Li et al., 2016). The sea-ice component of CESM2





is CICE version 5.1.2 (CICE5; Hunke et al., 2015) which features new mushy-layer thermodynamics (Turner and Hunke, 2015) with prognostic sea-ice salinity and an updated melt pond parameterization (Hunke et al., 2013). CICE5 uses the same horizontal mesh grid as the POP2 configuration to which it is coupled. For a more detailed description of the CESM2 ocean and sea-ice components, and changes relative to CESM1, the reader is referred to Danabasoglu et al. (2020). The initial conditions are derived from WOA13 (Locarnini et al., 2013; Zweng et al., 2013).

NCAR's low-resolution configuration (NCAR-L) utilizes a dipole mesh grid with the grid northern pole displaced into Greenland. The horizontal resolution (nominal 1°) is uniform in the zonal direction (1.125°), but varies in the meridional direction (from 0.27° at the Equator to ~0.5° at mid-latitudes). The z-coordinate vertical grid has 60 levels, going from 10 m at the surface to 250 m in the deep ocean to a maximum depth of 6000 m. The sub-gridscale closures and parameter settings used in NCAR-L are well-documented (Danabasoglu et al., 2012, 2014, 2020); some of the details are listed here to facilitate model intercomparison. NCAR-L employs the skew-flux form of the GM isopycnal transport parameterization (Griffies, 1998), with depth-dependent thickness and isopycnal diffusivities (Ferreira et al., 2005; Danabasoglu and Marshall, 2007) from roughly 3000 m2 s-1 in the near surface to 300 m2 s-1 in the deep ocean. Near surface mesoscale diabatic fluxes are also parameterized (Ferrari et al., 2008; Danabasoglu et al., 2008) with diffusivity set to 3000 m2 s-1, while the near-surface restratification effects of submesoscale mixed layer eddies are parameterized following Fox-Kemper et al. (2008, 2011). A modified version of the KPP parameterization (Large et al., 1994; Danabasoglu et al., 2006) is used for vertical mixing, with non-uniform background diffusivity that reflects tidal mixing effects (Jayne, 2009). NCAR-L (but not NCAR-H) uses an overflow parameterization to represent the density-driven flows of the Denmark Strait, Faroe Bank Channel, and the Weddell Sea (Danabasoglu et al., 2010). NCAR-L uses hourly coupling rather than the daily coupling used in previous CORE publications (e.g., Danabasoglu et al., 2014).

NCAR's high-resolution configuration (NCAR-H) is based on versions documented in McClean et al. (2011) and Small et al. (2014), but it has been updated to the CESM2 code base. It utilizes a tripole grid with the grid northern poles in North America and Asia. The horizontal grid (nominal 0.1°) varies from 11 km at the Equator to 2.5 km at high latitudes, and the vertical grid (62 levels) is the same as that used in NCAR-L but extends deeper to 6500 m with 2 additional levels for a total of 62 levels. NCAR-H (but not NCAR-L) uses a partial bottom cell formulation of the vertical grid for more accurate representation of bathymetry. In NCAR-H, biharmonic horizontal mixing is used for tracers and momentum, but there is otherwise no parameterization of eddy-induced mixing. New features in the CESM2 version of NCAR-H include the use of half-hour coupling and a modified virtual salt flux formulation that uses a local reference salinity (the estuary parameterization used in NCAR-L is not used in NCAR-H, but the latter does use new methods for redistributing continental freshwater fluxes over several vertical layers near the surface). In both NCAR-H and NCAR-L simulations, the surface stress is a function of ocean surface velocity (relative wind stress), and sea surface salinity is restored to WOA13 monthly climatology with a piston velocity of 50 m over one year. Both configurations use a precipitation factor, computed once per year, to prevent salinity drift as discussed in Appendix C of Danabasoglu et al. (2014).





### 2.3 AWI-FESOM

The Finite Element/volume Sea-ice Ocean Model (FESOM) configuration of the Alfred Wegener Institute Climate Model (AWI-CM, Sidorenko et al., 2015, 2018; Rackow et al., 2018, 2019; Sein et al., 2018) is also examined. Both the ocean and sea ice modules work on unstructured triangular meshes (Danilov et al., 2004; Wang et al., 2008), thus allowing for multi-
resolution simulations. FESOM version 1.4 (Wang et al., 2014; Danilov et al., 2015) is employed in this study and all the CMIP6 simulations as well. The tracer equations employ a flux-corrected advection scheme, as well as the KPP scheme (Large et al., 1994) for vertical mixing. The background vertical diffusivity is latitude and depth dependent (Wang et al., 2014). Mesoscale eddies are parameterized by using along-isopycnal mixing (Redi, 1982) and Gent-McWilliams advection (Gent and McWilliams, 1990) with vertically varying diffusivity as implemented in Danabasoglu et al. (2008). The eddy parameterization
is switched on where the first baroclinic Rossby radius is not resolved by local grid size. In the momentum equation the Smagorinsky (1963) viscosity in a biharmonic form is applied. The sea ice thermodynamics follow Parkinson and Washington (1979), with a prognostic snow layer to account for snow to ice conversion. The Semtner (1976) zero-layer approach, assuming linear temperature profiles in both snow and sea ice, is used in this model version. The elastic-viscous-plastic (EVP, Hunke and Dukowicz, 1997) rheology is used with modifications that improve the convergence (Danilov et al., 2015, Wang et al.,
2016). Sea surface salinity (SSS) is restored to monthly WOA13 climatology with a bolus velocity of 50 m over 300 days everywhere. The simulations are driven with the JRA55 forcing following the OMIP protocol. The air-sea turbulence fluxes are calculated using the bulk formulation of Large and Yeager (2009). The full ocean surface velocity is used in the calculation of wind stress (relative wind stress). The initial conditions are derived from PHC3.0 (Steele et al., 2001).

In the low-resolution setup, the bulk horizontal resolution is nominal 1 degree, with the North Atlantic sub-polar gyre
region and Arctic Ocean set to 25 km (see Figure 1a). Near equatorial resolution is 1/3°. This low-resolution grid has been used in previous CORE-II simulations (e.g., Wang et al., 2016b). The high-resolution grid was introduced in Sein et al. (2016). As the variability of sea surface height (SSH) can manifest the variability of mesoscale eddies, the horizontal resolution is scaled with the strength of the observed SSH variability on this grid. In particular, the resolution is about 10 km along the western boundary currents, the Antarctic Circumpolar Current, and the Agulhas Current region (Figure 1b). Before generating
this grid, the field of SSH variance is smoothed spatially to make sure that the main currents are within high resolution regions even if their positions change to some extent. The resolution is also increased along the coast and where the observed sea ice concentration variability is high. This multi-resolution grid has about 1.3M surface nodes, similar to the size of a uniform 1/4° mesh. In both setups, 46 z-levels are used, with 10 m layers in the upper 100 m.

**Figure 1. Horizontal resolution (km) of the two FESOM grids: (a) Low resolution, (b) High resolution.**

## 2.4 IAP-LICOM

LICOM (LASG/IAP Climate system Ocean Model) is a global ocean general circulation model (Zhang et al. 1989; Liu et al., 2004; Liu et al., 2012; Yu et al., 2018; Lin et al., 2020) developed by the Institute of Atmospheric Physics (IAP) from the Laboratory of Atmospheric Sciences and Geophysical Fluid Dynamics (LASG) of the Chinese Academy of Sciences (CAS),. LICOM is also the ocean component of both Flexible Global Ocean–Atmosphere–Land System model (FGOALS, e.g., Li et



al., 2013, Bao et al., 2013) and CAS Earth System Model (CAS-ESM, M. Zhang, private communication). LICOM version 3 (LICOM3) coupled with Community Ice Code version 4 (CICE4) through the NCAR flux coupler 7 (Craig et al., 2012; Lin et al., 2016) are employed for both low- and high-resolution experiments following the OMIP protocol. However, only the
thermodynamic part of CICE4 (no dynamics) was applied for high resolution. The surface salinity is restored to the monthly PHC3.0 climatology over the entire domain with a salinity piston velocity of 50 m /4 years (50 m/30 days for the sea ice regions). The full ocean surface velocity is used in the calculation of wind stress (relative wind stress).

LICOM3 is an ocean model with free sea surface. It uses the primitive equations with Boussinesq and hydrostatic approximations and is configured with the Murray's (1996) tripolar grid with two North "poles" at 65°N, 65°E and 65°N,
115°W for the low-resolution experiment and at 55°N, 95°E and 55°N, 85°W for the high-resolution experiment. The horizontal grid is the Arakawa B-grid with a resolution of approximately 1 degree in both zonal and meridional directions for the low-resolution experiment and 1/10° for the high-resolution experiment (11 km zonally and varying from 11 km at equator to 8 km in mid-latitudes). The vertical grid uses the eta-coordinate (Mesinger and Janjic, 1985) with 30 and 55 levels for the low- and high-resolution experiments, respectively. The low-resolution and high-resolution LICOM3 have a total of 360×218
and 3600×2302 grid points in the horizontal, respectively. The two-step preserving shape advection scheme (Yu, 1994; Xiao, 2006) and the implicit vertical viscosity/diffusivity (Yu et al., 2018) were adopted for both momentum and tracer equations. The split-explicit Leapfrog with Asselin filter is used for the time integration of both momentum and tracer.

The vertical viscosity and diffusion coefficients in the mixed layer have computed by the scheme of Canuto et al. (2001, 2002) with the background values of 2 $10^{-6}$ $m^2$/s and the upper limit of 2 $10^{-2}$ $m^2$/s. Recently, a tidal mixing scheme of St.
Laurent et al. (2002) has been adopted in LICOM3 by Yu et al. (2017). The Laplacian form with the coefficient of 5400 $m^2$/s was chosen for the horizontal viscosity in the low-resolution version and the biharmonic form with the coefficient of -2.8 $10^{10}$ $m^4$/s for the high-resolution experiment. In the low-resolution LICOM3, the isopycnal tracer diffusion scheme of Redi (1982) and the eddy-induced tracer transport scheme of Gent and McWilliams (1990, GM) with the same coefficients are used to parameterize the effects of mesoscale eddies on the large-scale circulation. Two tapering factors of Large et al. (1997) and a
buoyancy frequency ($N^2$) related thickness diffusivity of Ferreira et al. (2005) are also employed. However, for the high resolution, GM scheme has been turned off and the biharmonic form of the isopycnal diffusivity with the coefficient of -2.8 $10^{10}$ $m^4$/s was applied. Besides, the chlorophyll-a dependent solar penetration of Ohlmann (2003) introduced by Lin et al. (2007) was also implemented in both simulations.

## 3 Models time evolution and drift

### 3.1 Mean kinetic energy

Figure 2 shows the evolution of the domain-averaged mean kinetic energy for all experiments (solid lines for the high-resolution experiments, dashed lines for low resolution experiments) from 1958 to 2018. The evolution is very similar between different models, all exhibit a quick spin-up of the kinetic energy in the first five years which levels off for the rest of the



integration. Not surprisingly, the total kinetic energy is significantly higher for the high-resolution experiments over the low-

resolution experiments. For the high-resolution configurations, the FSU-HYCOM has the highest kinetic energy, with a globally averaged value of ~35 $10^{-4}$ $m^2/s^2$ and the IAP-LICOM has the lowest kinetic energy, with a globally averaged value of ~15 $10^{-4}$ $m^2/s^2$ in the high-resolution configuration. For comparison, a previous high-resolution 1/10° global simulation, performed with an older version of POP by Maltrud and McClean (2005) using daily NCEP/NCAR reanalysis forcing and absolute winds in wind stress calculations, has a global averaged kinetic energy at 25-30 $10^{-4}$ $m^2/s^2$ (see their Figure 1). The

higher kinetic energy in FSU-HYCOM can be partially explained by the wind stress formulation which does not take into account the ocean current velocities (absolute winds) while the other three models do (relative winds). The latter has an eddy killing effect that can reduce the total kinetic energy by as much as 30% (see Renault et al., 2019, for a review). This is roughly the difference that is seen between FSU-HYCOM and NCAR-POP and POP with absolute winds in the wind stress (Maltrud and McClean, 2005) has a level of kinetic energy that is close to FSU-HYCOM. But even with the highest resolution used here

(~0.1°), the total kinetic energy remains significantly lower that what can be inferred from observations and higher resolution models (closer to 50 $10^{-4}$ $m^2/s^2$, i.e. Chassignet and Xu (2017)). The increase in total kinetic energy from the low- to the high-resolution configuration is approximately a factor of 4 for all models, except for AWI-FESOM (factor 2 only). This is probably because the high-resolution AWI-FESOM has a variable grid spacing (Figure 1b) and does not resolve the Rossby radius of deformation everywhere.

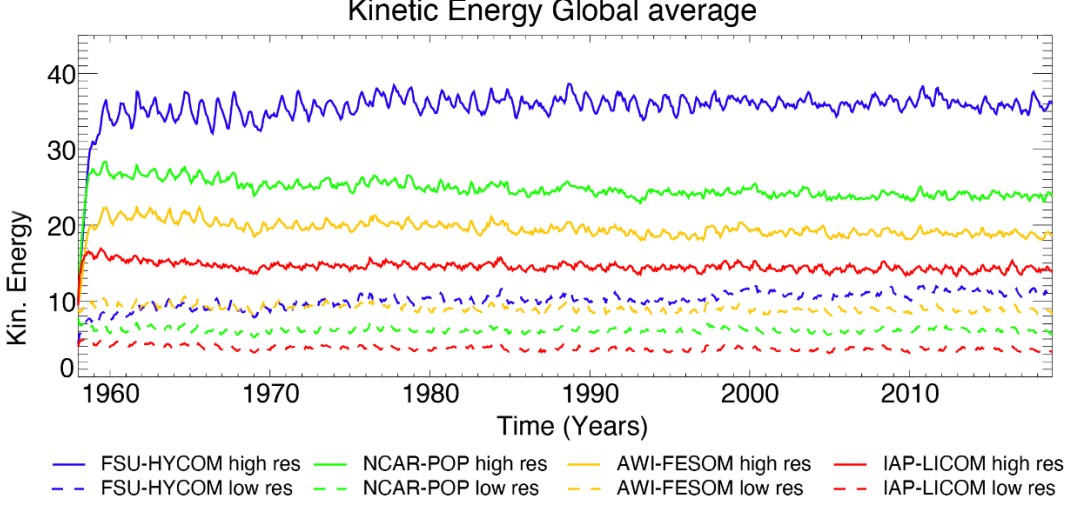


**Figure 2: Time evolution of the domain-averaged kinetic energy ($10^{-4}$ $m^2/s^2$) for all experiments.**

## 3.2 Global mean sea level, temperature, and salinity

As stated by Griffies et al. (2014), "[t]he CORE [and OMIP] protocols (Griffies et al., 2009; Danabasoglu et al., 2014; Griffies et al., 2016) introduce a negligible change to the liquid ocean mass (non-Boussinesq) or volume (Boussinesq), and the

salt should remain nearly constant (except for relatively small exchanges with the sea ice). Changes to the simulated global





mean sea level should arise only because of thermosteric effects (i.e., changes in ocean heat content and redistribution of heat) in simulations that preserve salt content (i.e., that have zero net surface freshwater flux). In most models, the global sea level time evolution (Figure 3) is dominated by changes in the global mean ocean temperature (Figure 4a). IAP-LICOM is the exception, in which the global sea level shows a downward trend until ~1990, and then slowly rises. This is due to an increase

in global mean salinity (Figure 4b) which dominates the global sea level changes despite a large increase in global volume mean temperature (Figure 4a). This increase is explained by the lack of surface restoring salt-flux normalization. The salt flux normalization is given in Appendix B.3 of Griffies et al. (2009) and Appendix C in Danabasoglu et al. (2014), which ensure there is no net salt added to or removed from the ocean-sea ice system (Griffies et al., 2014). One can also note that most models show an increase in global temperature and sea surface height after 1980-90 associated with rising air temperature.

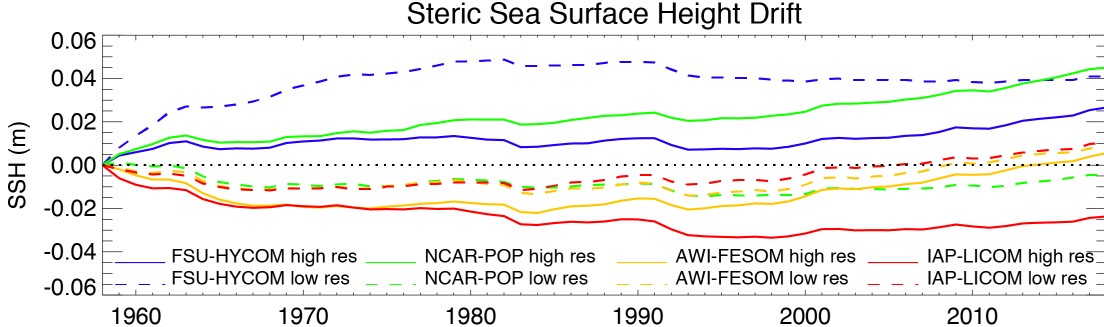

**Figure 3: Time evolution of the global steric sea surface height change for all experiments.**

An increase in the horizontal resolution does not necessarily imply a reduction in temperature and salinity drift and no coherent picture emerges from the comparison. If one focuses on the time evolution of the globally averaged temperature in the upper 700 m (Figure 4c), there are only small changes for AWI-FESOM and IAP-LICOM, while FSU-HYCOM shows a

significant reduction in the drift and NCAR-POP an increase as resolution is increased. For salinity in the upper 700 m (Figure 4d), the increase in resolution significantly reduces the drift in NCAR-POP and FSU-HYCOM, no changes for IAP-LICOM, and a significant increase for AWI-FESOM. It is important to note that, while the salt flux restoring may differ among the four models, it remains the same for each model as the resolution is increased. When considering the upper 2000 m (Figure 4f), there is a significant reduction in the global salinity drift for NCAR-POP and FSU-HYCOM, less so for IAP-LICOM, and no

changes for AWI-FESOM. We note that all high-resolution models warm faster over the upper 2000 m and global temperature than their lower-resolution partners, which is not true for the upper 700 m. Figure 5 shows in more detail the evolution of the global temperature and salinity as a function of depth. AWI-FESOM shows the smallest changes in temperature throughout the water column, but the largest in salinity despite having a salt flux adjustment to ensure that the global salinity remains constant (shown in Figure 2). There is a significant freshening in the upper 400 m compensated by a salinification in the deeper

ocean. This is more pronounced in the high-resolution experiment. Increasing the resolution significantly improves the drift in both temperature and salinity in FSU-HYCOM, but not so much in the other simulations. One could actually argue that the



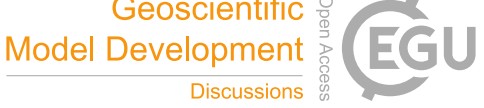

drift is stronger in NCAR-POP with a significant warming in the upper 400 m and freshening in the upper 100 m. In the next section, we investigate in more details the evolution of the temperature and salinity as function of depth and time by ocean basin.

**Figure 4: Time evolution of the global temperature (°C) and salinity (psu) change for all experiments (depth-integrated, upper 700 m, upper 2000 m).**



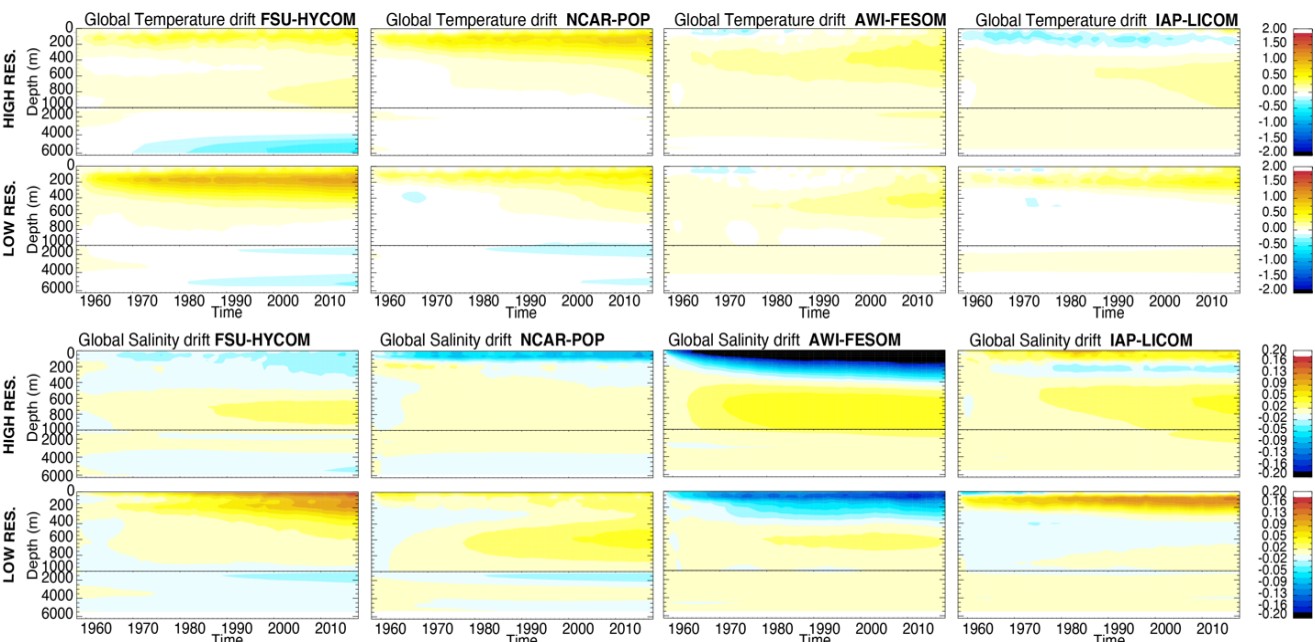

**Figure 5: Time evolution of the global temperature (°C) and salinity (psu) changes as a function of depth for all experiments.**


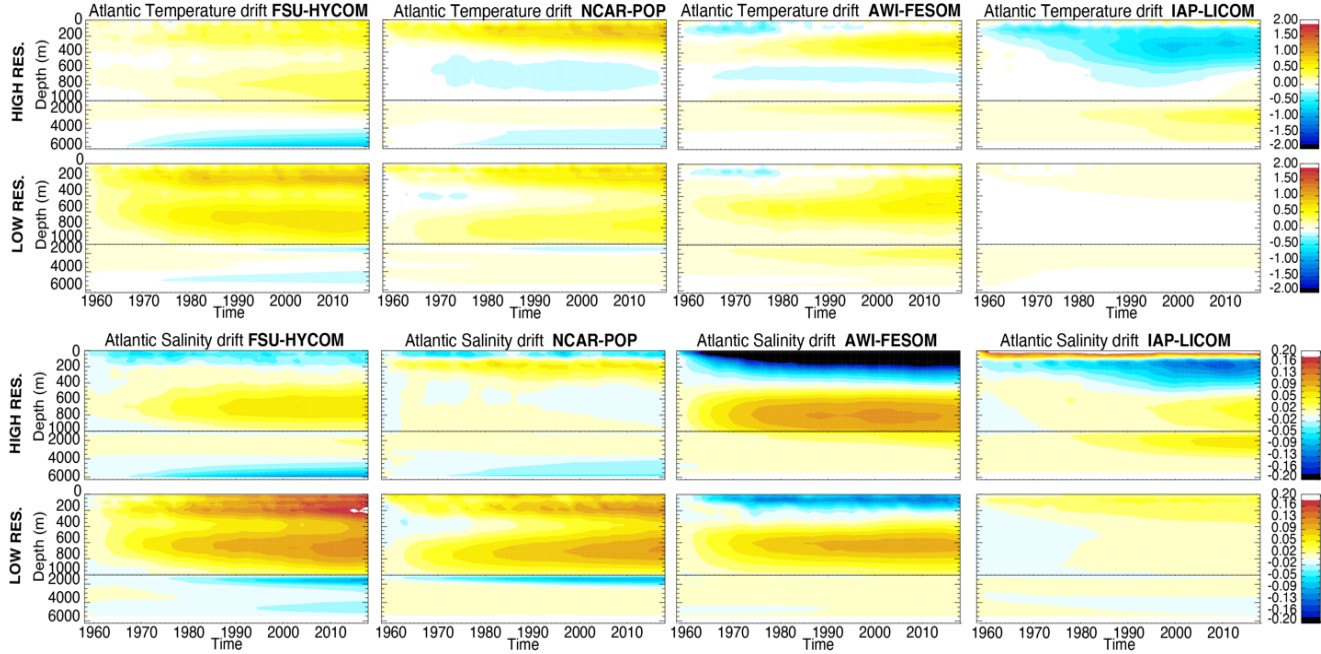

**Figure 6: Time evolution of the Atlantic Ocean temperature (°C) and salinity (psu) changes as a function of depth for all experiments.**

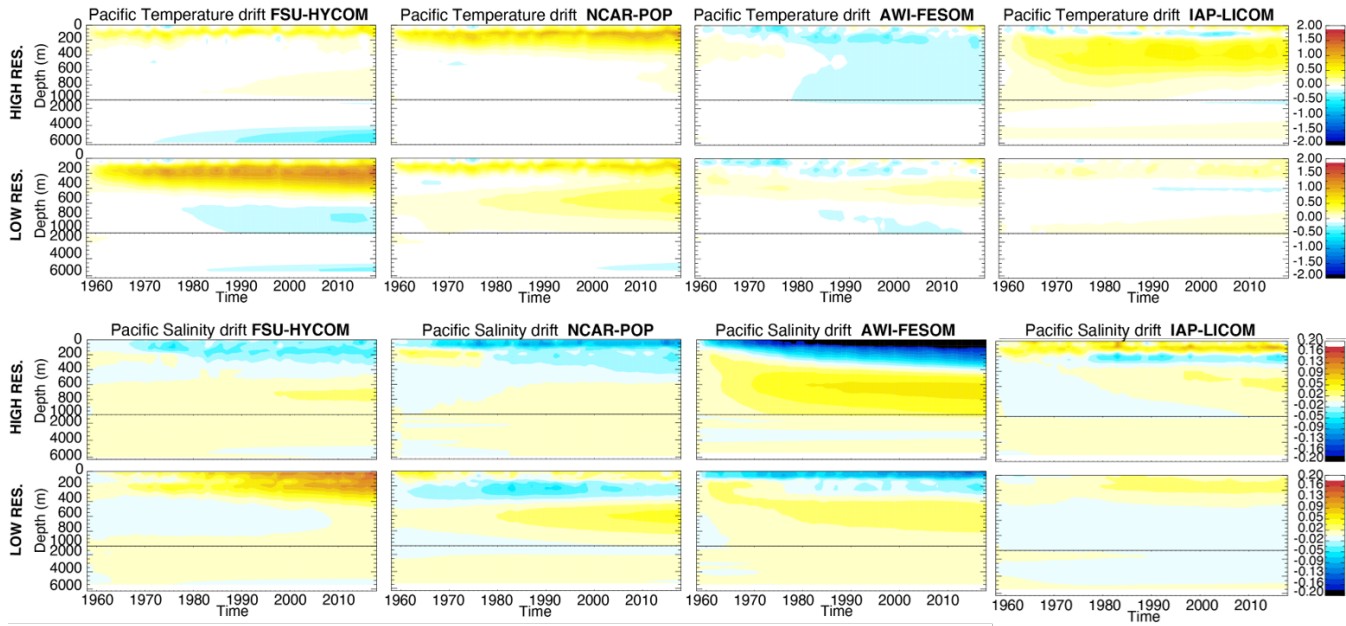

**Figure 7: Time evolution of the Pacific Ocean temperature (°C) and salinity (psu) changes as a function of depth for all experiments.**


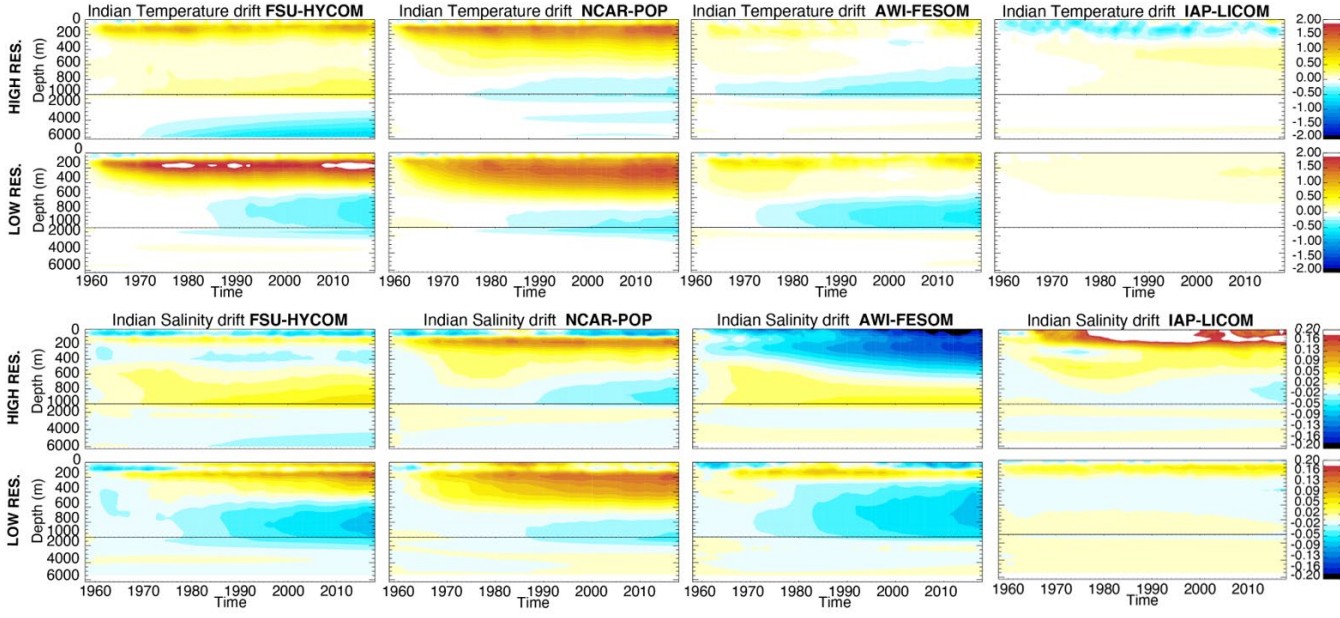

**Figure 8: Time evolution of the Indian Ocean temperature (°C) and salinity (psu) changes as a function of depth for all experiments.**



### 3.3 Temperature and salinity bias (depth vs. time) by ocean basin

Figures 6, 7, and 8 show the time evolutions of the horizontal-mean depth profiles of the temperature and salinity for the Atlantic, Pacific, and Indian Oceans, respectively. The color bar is the same in all figures, including Figure 5 (global), therefore allowing for a qualitative estimate of where the drift is most significant. To a large extent, the time evolution in each of the major ocean basins mimics that of the global, but with some significant differences. For the Atlantic Ocean (Figure 6), there is a surface freshening in the upper 100 to 200 m as resolution is increased in FSU-HYCOM, NCAR-POP, and AWI-FESOM.

IAP-LICOM, on the other hand, becomes more saline and warmer in the upper 100 m. The latter is accompanied by a large freshening and cooling below 100 m to approximately 600 m. Overall, the drift is smaller in the Pacific for most models, with a freshening in the upper ocean for all models as resolution is increased. The exception is again IAP-LICOM which shows a significant increase in salinity in the upper 200 m and this could be a consequence of the fact that there is no zero normalization of the surface restoring salt-flux. The temperature bias in the Pacific Ocean is similar to that of the global ocean. In the Indian

Ocean (Figure 8), FSU-HYCOM and NCAR-POP exhibit larger biases than AWI-FESOM and IAP-LICOM, but those biases are smaller at the higher resolution.

### 3.4 Model mean temperature and salinity versus initial climatology

The drift plots from section 2b,c indicate that temperature and salinity bias structures are well established within the first two decades of the spin up so that time averages computed over the latter decades of the simulations should provide a

reasonable estimate of the stationary biases characterizing each model. Figure 9 shows latitude-longitude maps of surface temperature and salinity biases computed over the final 10 years of integration (historical years 2009-2018) with respect to the climatology used to initialized the models (GDEM, WOA13, or PHC3.0, see section 2 for details). With the possible exception of AWI-FESOM, all models exhibit clear reductions in SST bias as resolution is increased. The largest bias reductions are seen in the Southern Ocean, in the Agulhas retroflection region, and along the Gulf Stream extension in the North Atlantic.

All these changes are presumably primarily related to an improved representation of the pathways of strong surface currents in those regions, which could result not only from differences between resolved and parameterized eddies but also from differences in resolved bathymetry. SST bias in eastern boundary upwelling regions has been shown to be strongly sensitive to atmospheric forcing resolution (Tsujino et al., 2020), somewhat sensitive to the regridding technique used for near-surface winds (Small et al., 2015), and also slightly sensitive to ocean resolution for seasonal means (Kurian et al., 2020). However,

Figure 9 shows minimal change in annual mean SST bias in the upwelling zones off the west coasts of Namibia, Peru, and North America. In addition to the Gulf Stream, all models exhibit some degree of bias reduction in the Brazil-Malvinas Confluence zone at high resolution, but no such systematic improvement is evident in the Kuroshio-Oyashio extension region. Sea surface salinity bias is generally reduced globally in the high-resolution simulations, with the exception again being AWI-FESOM which exhibits a more pronounced negative salinity bias in line with the enhanced near-surface salinity drift at high

resolution in that model (cf. Figures 5-8).

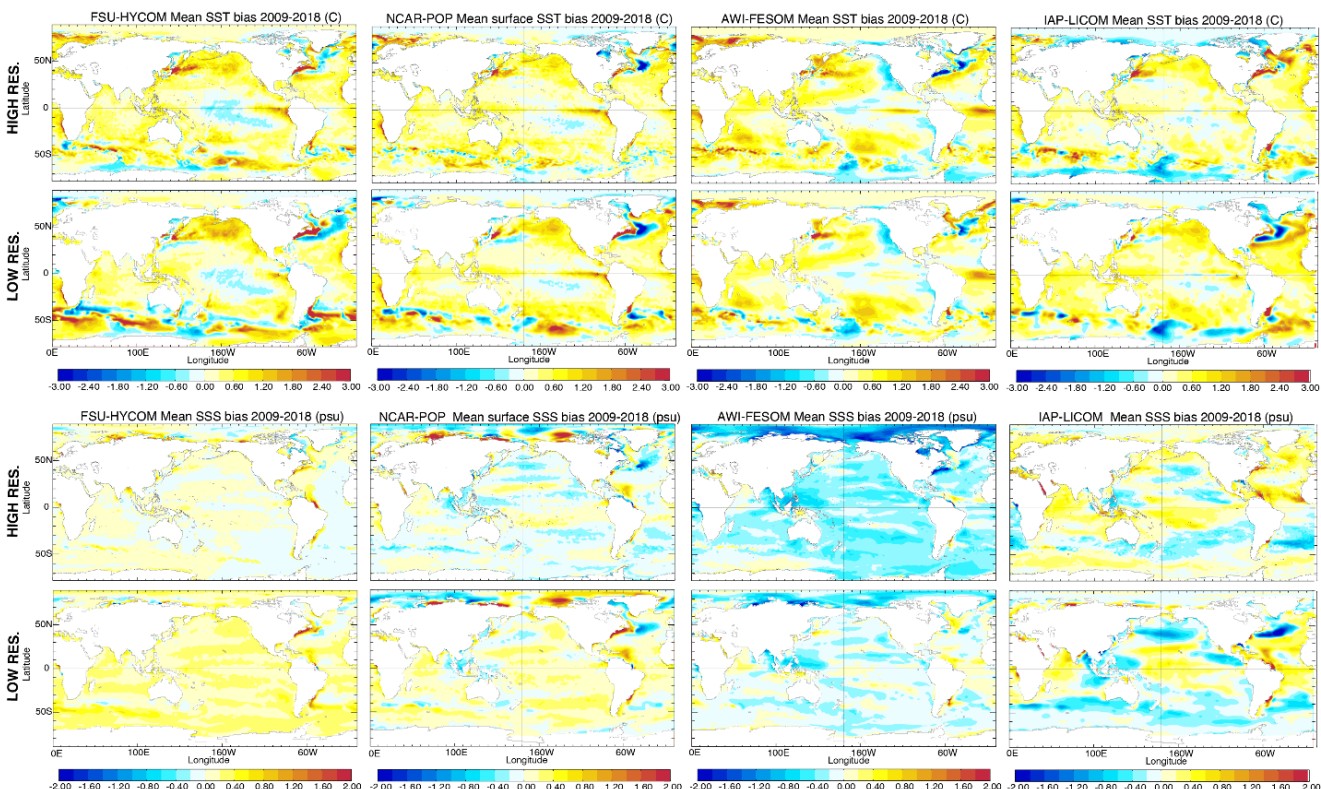

**Figure 9: Modeled surface temperature (°C) and salinity (psu) difference with the climatology used to initialize the model.**

The vertical structure of mean temperature and salinity bias is displayed as zonal-averages across the different ocean basins in Figures 10-12. This reveals that the near-surface global drift toward warming seen in FSU-HYCOM and NCAR-POP

is partly related to a degradation of the tropical thermocline in all basins. In FSU-HYCOM, higher resolution improves the situation, while in NCAR-POP, the tropical thermocline temperature bias actually gets worse with resolution. In both FSU-HYCOM and NCAR-POP, and to some extent IAP-LICOM, large extratropical/polar temperature biases associated with intermediate and deep waters decrease with enhanced resolution. In AWI-FESOM, both high and low latitude biases get worse. It turns out that the relatively low near-surface temperature drift seen in the AWI-FESOM model (Figures 5-8) is related to

large, but largely compensating, anomalies of opposite sign at each depth level. Cold/warm anomalies that develop in the Tropics and at high latitudes in that model at both resolutions tend to disappear in global or basin-wide means. Such compensation does not occur for salinity which is too fresh in the thermocline and too saline in deeper intermediate waters in AWI-FESOM, a characteristic that gets worse as resolution is increased. The IAP-LICOM model exhibits some large changes in the sign and structure of zonal mean bias as resolution changes, but as with AWI-FESOM, it does not lend strong support

to the hypothesis that model temperature and salinity bias can be reduced by increasing model horizontal resolution. As already mentioned, the NCAR-POP model exhibits improved representation of high latitude intermediate and deep waters, but degraded representation of the tropical thermocline, as resolution is enhanced. FSU-HYCOM is the only model that shows





near ubiquitous reduction in temperature/salinity bias in all ocean basins in the high-resolution version; for the other models, greatly enhanced horizontal resolution does not deliver unambiguous bias improvement in all regions. This is a rather unexpected result.


**Figure 10: Global zonal temperature (°C) and salinity (psu) difference with the climatology used to initialize the model.**





**Figure 11: Atlantic zonal temperature (°C) and salinity (psu) difference with the climatology used to initialize the model.**





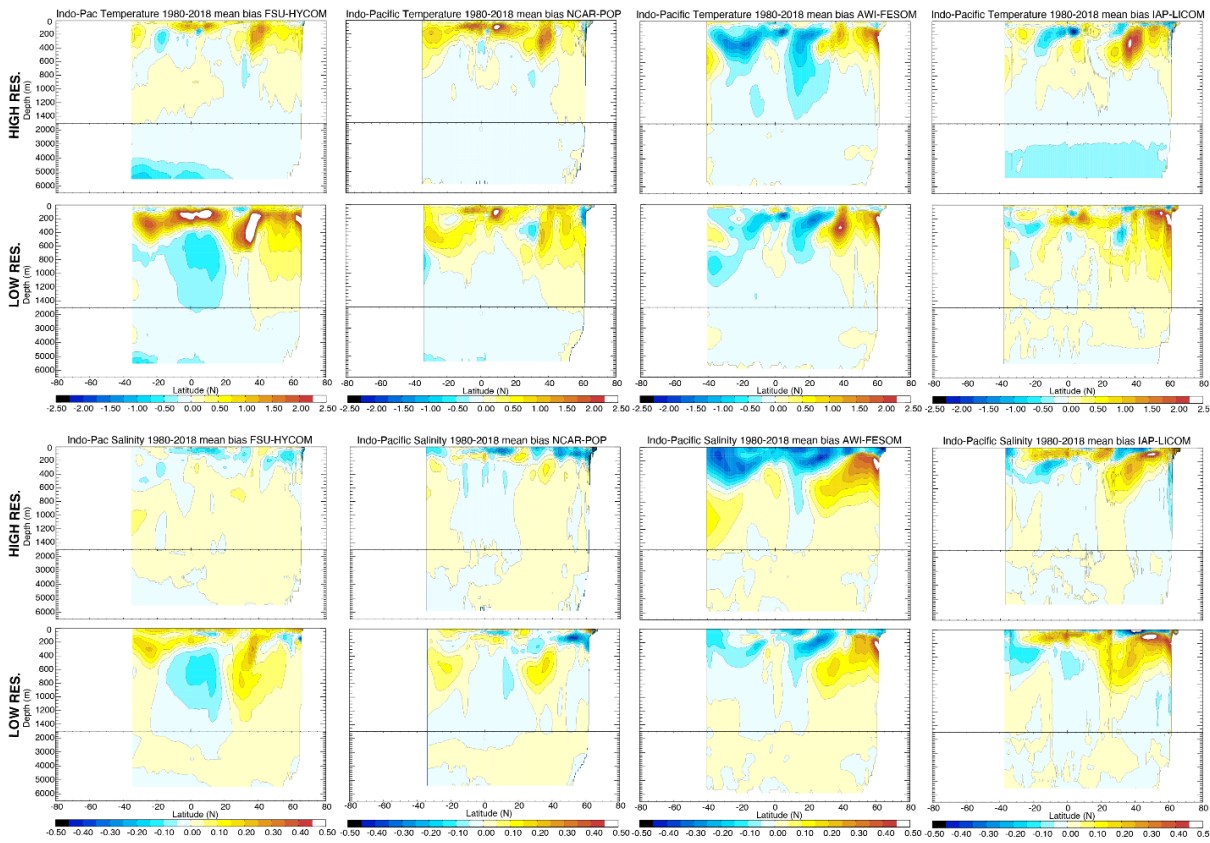

**Figure 12: Indo-Pacific zonal temperature (°C) and salinity (psu) difference with the climatology used to initialize the model.**

### 3.5 Northern and southern hemisphere sea-ice volume and extent

Sea ice is usually quantified and monitored by satellite in terms of sea ice extent, defined as the area with 15% or higher

sea-ice concentration. Figures 13a-b display the modeled (annual mean) northern and southern hemisphere sea-ice extent for all simulations and are compared to the latest observations from the National Snow and Ice Data Center (NSIDC, Fetterer et al., 2017). Despite a wide range of time mean sea-ice extent, all simulations represent well the observed variability of the sea-ice extent from 1979 to 2018. In the northern hemisphere, the sea-ice extent shows a clear decline since the beginning of the monitored period, whereas, in the southern hemisphere, the sea ice extent remains more or less stable until the last few years

of the integration with a small upward trend. Only the low-resolution version of FSU-HYCOM shows an inconsistent negative trend in the southern hemisphere sea-ice extent. It is known that the observed sea-ice extent is highly correlated with the near-surface air temperature (e.g., Olonscheck et al., 2019). Thus, the consistency of the temporal variability between different simulations and observations suggests that the air-temperature in the JRA55-do is quite realistic. It is also notable that in the northern hemisphere, all of the high-resolution models show less bias in sea-ice extent than their low-resolution counterparts.

In the southern hemisphere, all of the high-resolution models show less bias in extent than their low-resolution counterparts, with the exception of IAP-LICOM.



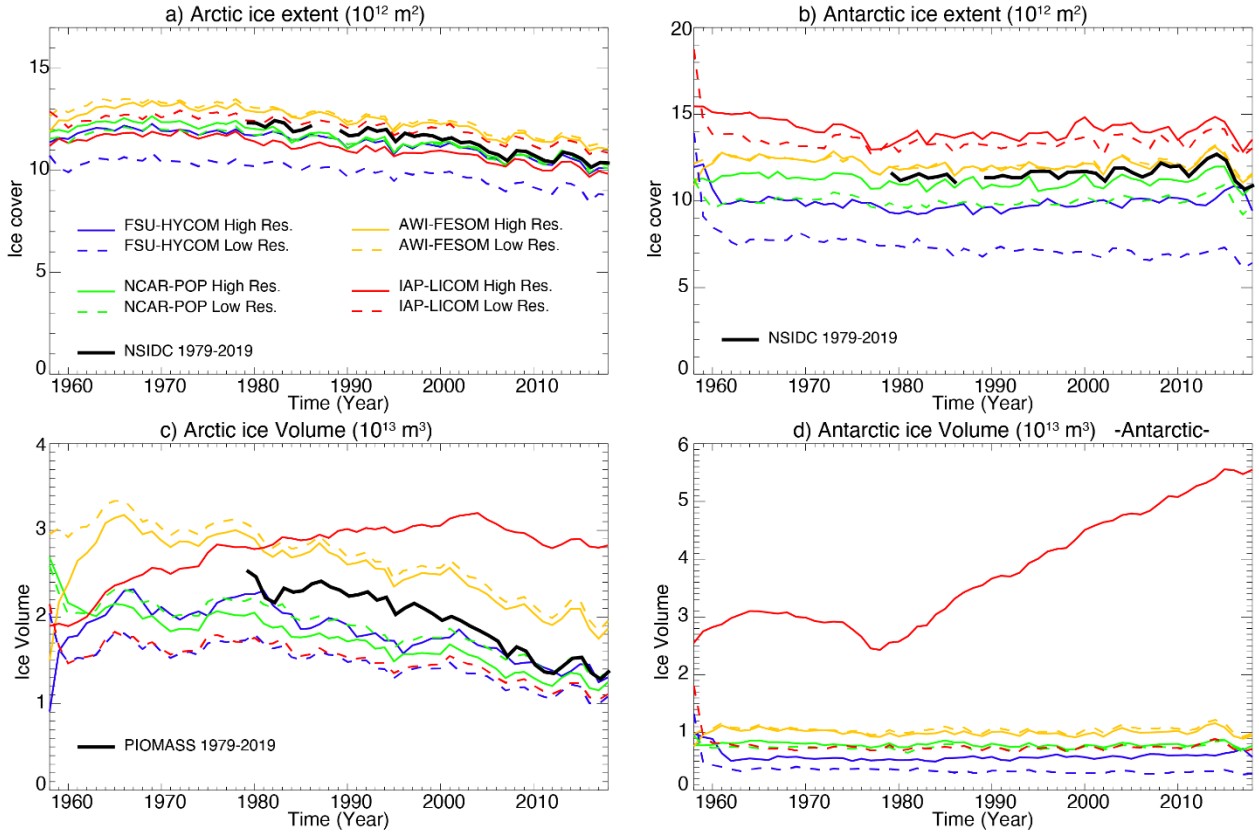

**Figure 13: Modeled time evolution of annual mean Arctic and Antarctic a-b) sea ice extent in $10^{12}$ m$^2$) and c-d) sea ice volume in $10^{13}$m$^3$. The black lines are observations from National Snow and Ice Data Center (NSIDC) and results from Pan-Arctic Ice Ocean**
**Modeling and Assimilating system (PIOMAS).**

Another measure of the sea-ice simulation performance is the sea-ice volume. As there are no reliable long-term sea ice volume observations, we use the model result from the Pan-Arctic Ice Ocean Modeling and Assimilating system (PIOMAS, Schweiger et al., 2011) at the Polar Science Center to assess the northern hemisphere sea-ice volume in our simulations. The modeled time evolution of the northern/southern hemisphere sea-ice volume is displayed in Figures 13c-d. Despite differences

in the mean averaged values, all but one simulation exhibit a continuous decline of the northern hemisphere sea-ice volume that is comparable to the PIOMAS observations (Figure 13c) and this result is not significantly influenced by the model's horizontal resolution. The outlier simulation is the high-resolution IAP-LICOM which shows an overall increase of northern hemisphere sea-ice volume (despite a decline in sea ice extent). This because its sea-ice model has only thermodynamic processes and the absence of dynamic sea-ice leads to an excessive accumulation and volume. For the southern hemisphere,

the time evolution of the modeled sea-ice volume is also similar for all simulations, except again for the high-resolution IAP-LICOM.





### 3.6 Meridional overturning circulation

The Atlantic meridional overturning circulation (AMOC) is often quantified by a meridional overturning streamfunction with respect to depth, $\psi_z$ at each latitude, defined as the meridional transport (in Sv) across the basin above a constant depth $z$:

$$\psi_z(y,z) = \overline{\iint_z^0 v(x',y,z',t)\,dz'dx'},$$

where $v$ is the 4-d meridional velocity, the overbar denotes a time average (here annual mean), and the $x$ integration covers the entire span of the Atlantic basin. The magnitude of the AMOC, or the AMOC transport, is then defined as the maximum of the streamfunction $\psi_z$ with respect to $z$, representing the total northward transport above the overturning depth (approximately 1000 m in the Atlantic Ocean). This important measure of the AMOC is quantified and monitored by the RAPID moorings near 26.5°N (e.g., Smeed et al., 2018).

The modeled annual mean AMOC transport across 26.5°N from 1958 to 2018 is displayed in Figure 14a, along with the RAPID results (2005-2017). The vertical distribution of the time mean zonal streamfunction in the Atlantic Ocean is shown later in Figure 25. For the four models, the high-resolution simulations have a higher AMOC transport than their low-resolution counterparts: the mean AMOC transport in 2004-2018 is 7.8-14.9 Sv in the four low resolution simulations, versus 14.0-19.8 Sv in the four high-resolution (17.2 ±1.5 Sv as observed by RAPID, e.g. Smeed et al., 2018). However, there is not an obvious difference in the AMOC sensitivity to forcing or in the trend between the low- and high-resolution experiments. The simulated time evolution of the AMOC transports at this latitude depends on the model, its parameterizations, resolution, and on the number of spin-up cycles (Danabasoglu et al., 2016). Despite the fact that only the simulations were only integrated over one JRA55-do cycle (1958-2018), all four low-resolution models show a similar multi-decadal variability with a transport decrease from 1958 to late 1970s, an increase from the late 1970s to late 1990s, and a decrease again thereafter. This multi-decadal variability is also present in the CORE-II simulations (Danabasoglu et al., 2016; Xu et al., 2019). In the high-resolution simulations, FSU-HYCOM, NCAR-POP, and AWI-FESOM exhibit a similar multi-decadal variability (of low transports in the late 1970s and high transports in the early 1990s) as seen in previous basin-scale simulations (e.g., Böning et al., 2006; Xu et al., 2013). The decline of the AMOC transport from early 1990s to 2000s may be a consequence of the warming and reduced deep convection in the western subpolar North Atlantic that has been documented quite extensively (e.g., Hakkinen and Rhines, 2004; Yashayaev, 2007). The high-resolution AWI-FESOM model has a strong AMOC decreasing trend (of 4-5 Sv) initially during the first 20 years and then has a time evolution of the AMOC that is very similar to that of FSU-HYCOM and NCAR-POP. The high-resolution IAP-LICOM shows an increasing AMOC transport from 1958 to the early 1990s (by 10 Sv) and decrease thereafter (by 5 Sv), which is quite different from the other three models, but it does appear to capture the increase over the last few years.





**Figure 14: Modeled annual mean Atlantic meridional overturning circulation (AMOC) transports at a) 26.5°N and b) 34°S; and c) global Meridional overturning circulation (MOC) transports at 34°S. Solid back line in panel a) are results from RAPID array.**




The AMOC transport in the South Atlantic Ocean has been quantified near 34°S, using several observational techniques including expendable Bathythermograph (XBT), Argo profiles, and moored current meter arrays in the western and eastern boundaries (Baringer and Garzoli, 2007; Dong et al., 2009, 2014, 2015; Garzoli et al., 2013; Goes et al., 2015; Meinen et al., 2013; 2018) and these observations yield a time mean AMOC transport of about 14-20 Sv (see Xu et al. (2020) for an in-depth

discussion of the circulation in the South Atlantic Ocean). The modeled temporal evolution of the AMOC transports at this latitude are overall similar to that at 26.5°N in the North Atlantic (Figures 14a-b). The range of the modeled time mean AMOC transport at 34°S in the high-resolution simulations, from 14.7 Sv in FSU-HYCOM to 20.1 Sv in IAP-LICOM, is about the same as the observational range mentioned above.

The Antarctic Bottom Water (AABW) cell of the global overturning circulation (Talley, 2013) can be visualized as a

streamfunction similar to Equation 1, except that the zonal integration now spans across the full longitude circle. The transport associated with this cell is defined as the minimum of the streamfunction that is typically found near 3500 m (Lumpkin and Speer, 2007) and is a measure of the northward flow of the near bottom water (AABW and/or Circumpolar Deep Water, CDW) from the Southern Ocean into the Atlantic, Pacific, and Indian Oceans that eventually upwells and returns southward at a shallower depth. Several observational estimates are provided based on hydrographic sections near 32°S, but the uncertainty

remains quite high (29±7.6 Sv in Talley (2013)). Figure 14c displays the modelled transport at 34°S. The low-resolution simulations show a consistently lower transport, with the mean transport value for 2004-2018 ranging from 7.4 Sv in IAP to 15.3 Sv in FSU-HYCOM. The low transport in IAP-LICOM is due to a long-term downward trend in this simulation: from 20 Sv in the early 1960 to 5 Sv in the late 2010s. The other three models, especially FSU-HYCOM, exhibit relatively stable transports in the last 30 years (of the integration). The high-resolution simulations show a much wider range, with time mean

transports (2004-2018) ranging from 11.7 Sv in IAP-LICOM to 26.5 Sv in AWI-FESOM. The time evolution of the 34°S transports in the high-resolution simulations typically shows an increase in the early stage of the integration followed by a gradual decrease afterward. But the timescale of the increase and decrease periods varies significantly between models. As a result, the transport is relatively stable in IAP-LICOM and AWI-FESOM for the last 30 years of the integration but continues to decrease slowly in the NCAR-POP and FSU-HYCOM. Interestingly, despite the large difference in time mean transports,

all four high-resolution models exhibit a similar interannual variability at this latitude for the last 30 years of the integration.

### 3.7 Drake Passage and Indonesian Throughflow transports

The Antarctic Circumpolar Current (ACC) is a strong oceanic current that flows eastward around Antarctica. It connects the Pacific, Atlantic, and Indian Oceans to its north and is the primary means of inter-basin exchange, enabling a truly global overturning circulation (e.g., Gordon, 1986; Schmitz, 1996; Talley, 2013). An accurate knowledge of the ACC transport is

fundamental to understand its influence on the global circulation. Substantial observational efforts have been made toward quantifying the ACC transport, especially in Drake Passage, where the ACC is constricted between the Antarctica and the southern tip of the South America. The major observations include: 1) the International Southern Ocean Studies (ISOS) program in the late 1970s and early 1980s (Whitworth, 1983; Whitworth and Peterson, 1985); 2) the repeat hydrographic




sections along the World Ocean Circulation Experiment (WOCE) line SR1b (e.g., Cunningham et al., 2003); 3) the repeat

shipboard ADCP surveys that directly measure the current in the top 1000 m of the water column (Firing et al., 2011); 4) the

DRAKE program that combines a moored current meter array (2006-2009) and satellite observations (e.g., Koenig et al.,

2014), and 5) the cDrake program of a high-resolution bottom current meter mooring array and cPIES observations from 2007

to 2011 (e.g., Chidichimo et al., 2014; Donohue et al., 2016). The time mean, full-column transport estimates range from

134±15 Sv based on ISOS, to 141±2.7 Sv based on DRAKE program, and to 173.3±10.7 Sv based on the cDrake results.

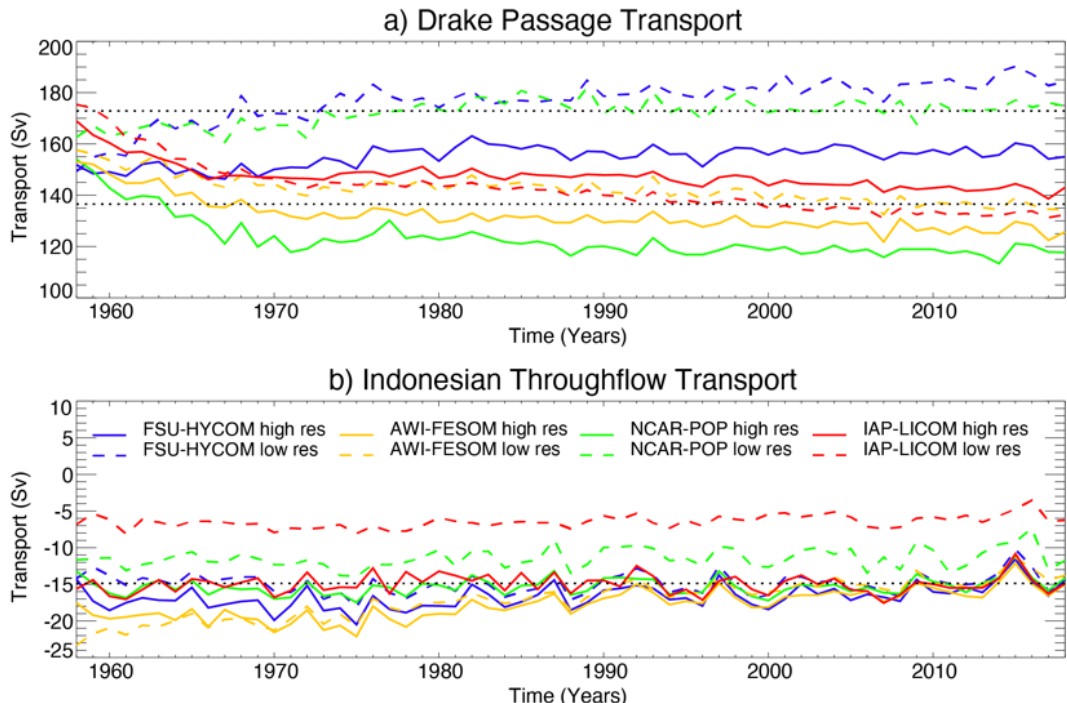


**Figure 15: Modeled annual mean volume transport of the ACC through Drake Passage and the Indonesian Throughflow (ITF) from Pacific to Indian Ocean. The dotted lines in a) is the observational range between the canonical transport value (134 Sv) based on the ISOS program and 173 Sv based on cDrake observations. The dotted line in b) is the observed ITF transport value (15 Sv) based on INSTANT observations.**

The modeled annual mean Drake Passage transport is displayed in Figure 15a together with the mean observational range

based on the most recent observations (134-173 Sv). The transport in the low-resolution simulations differ from each other's:

IAP-LICOM and AWI-FESOM show a continuous decrease; FSU-HYCOM shows show a continuous increase; whereas

NCAR-POP shows an increase in the first 20 years, then levels off for the next 40 years of the integration. It is expected that

the low-resolution models will be sensitive to different parameterizations and their ability to simulate eddy saturation (Gent

and Danabasoglu, 2011; Munday et al. 2013). As a result, the time mean transport for the last 15 years of these low-resolution

simulations are either near the edge or outside of the above observational range: 132, 135, 173, and 190 Sv, respectively, for

IAP-LICOM, AWI-FESOM, NCAR-POP, and FSU-HYCOM. The evolution of Drake Passage transport is slightly more

similar in high-resolution simulations (compared to the low-resolution counterparts). Even at high resolution, mean ACC



transport is sensitive to subgrid schemes (Pearson et al. 2017). Three models show a fast decrease in the first 10-15 years and
then a very small decrease thereafter, except that FSU-HYCOM shows a small increase in the first 20-years and then levels
off thereafter. Interannual variability among these models is more correlated than in the lower resolution models. The ACC
transports in NCAR-POP and AWI-FESOM, 120 and 130 Sv, respectively, are slightly lower than the mean transport estimates
from the latest observations, but still within the uncertainty range; while the transports in IAP-LICOM and FSU-HYCOM,
145 and 157 Sv, respectively, are within the mean observational range.

The long-term trend in the modeled ACC transport merits some further discussions. Observations indicate that, as shown
by a higher Southern Annular Mode index, the westerly winds in the Southern Ocean have been intensified since the 1970s.
This increase in westerly winds is also present in the JRA55-do forcing, yet none of the high-resolution simulations exhibit a
long-term upward trend (three models even have a slightly weakening trend). The ACC transport time series from DRAKE
program (Koenig et al., 2014) as well as from the WOCE line SR1b also show a stable ACC transport from early 1990s. Given
that the ACC is wind-driven, a lack of dependence for the ACC transport on the zonal wind at long-term timescale may be at
first surprising, but it has also been shown that the strengthening of winds does generate more eddy activity along the path of
the ACC, without necessarily changing its total transport (e.g., Hallberg and Gnanadesikan, 2006; Morrison and Hogg, 2013;
Munday et al., 2013; Farneti et al. 2015). Following this argument, one may expect to see an increasing ACC transport in low-
resolution, non-eddying simulations as long as the coefficient used in the eddy parameterization (i.e., GM) remains constant
(Gent and Danabasoglu, 2011). Only FSU-HYCOM shows a long-term increase in the ACC transport and two models (IAP-
LICOM, and AWI-FESOM) even show a long-term decrease. This is likely a consequence of the eddy parameterization choice
made by the different models, assuming that the ACC is quasi-equilibrated and that changes are not related to buoyancy
changes in the Southern Ocean (e.g., rate of bottom water formation). All models have weaker Drake Passage transports at
higher resolution, except for IAP-LICOM.

The Indonesian Throughflow Flow (ITF) provides another key inter-basin exchange mean of water mass as well as heat
and freshwater. It connects the tropical Pacific and Indian Oceans directly and, through the Agulhas Leakage, eventually feeds
into the upper AMOC limb in the South Atlantic Ocean as part of the global MOC scheme (e.g., Gordon, 1986; Schmitz, 1996;
Talley, 2013). A good overview of the Indonesian Sea oceanography and the ITF observations can be found in Gordon (2005)
and Gordon et al. (2010). The 3-year mean total ITF transport measured by the International Nusantara Stratification and
Transport (INSTANT) program in 2004-2006 is 15 ±2.5 Sv (Sprintall et al., 2009; Gordon et al., 2010), about 30% greater
than the values of non-simultaneous measurements made prior to 2000. The ITF transport variability exhibits a close
relationship to the phase of El Niño—Southern Oscillation (Meyers, 1996).

The modeled annual mean ITF transport is displayed in Figure 15b. The low-resolution simulations exhibit a wide range
of the ITF transports, ranging from about 5 Sv in the IAP model, to 11 Sv in NCAR-POP model, and to 15 Sv in FSU-HYCOM
and AWI-FESOM (30-year mean). The ITF transports exhibit a gradual decrease in AWI-FESOM, but are relatively stable in
the other three models. In the high-resolution simulations, however, the models yielded a quite similar ITF transport, especially
in the second half of the integration. All four models represented a similar mean transport of 15 Sv that is close to observations





and all show a very similar variability. While large-scale forcing mechanisms are thought to set the basic level of ITF transport (Godfrey, 1989), the difference between the low-resolution and high-resolution simulations is likely impacted by the model's
ability to represent the small-scale bathymetry feature of the Indonesian Seas (Jochum et al., 2009), after all, the ITF enters the Indian Ocean through a number of narrow straits that are hard to represent in the low-resolution models. IAP-LICOM and NCAR-POP have low ITF transports at low-resolution and appear to have stronger sensitivity to model resolution.

## 4 Stationary ocean climate

### 4.1 Sea Surface Height (SSH) and Eddy Kinetic Energy (EKE)

The impact on the circulation of increasing the horizontal resolution is two-fold. First, the solution becomes more nonlinear and allows for a better representation of western boundary currents. Second, the first Rossby radius of deformation is resolved over most of the globe (Hallberg, 2013) and eddies are formed through barotropic and baroclinic instabilities, although higher vertical modes including submesoscale eddies are not often resolved (nor are they resolved by altimetry, i.e. AVISO). Figure 16 displays the mean sea surface height bias with respect to AVISO and its 5-day variance over the 1993-
2018 period. Overall, the large-scale patterns are well represented globally, and one can clearly see the improved representation of the western boundary currents as resolution is increased (Gulf Stream and Kuroshio). In the North Atlantic, most coarse resolution models to date have the tendency to exhibit an overshooting Gulf Stream and a poor representation of the North Atlantic Current (NAC) at the Northwest Corner. This is the case for three out of the four models. Instead of flowing northeastward along the continental rise past the Flemish Cap and continuing northwestward as in the observations (Rossby,
1996), the North Atlantic Current is strongly zonal in NCAR-POP, AWI-FESOM, and IAP-LICOM, and does not turn northeastward at the Flemish Cap. This has been a long-standing issue for many ocean components of the CMIP climate models and it does not necessarily improve as the computational mesh is refined. Increasing the horizontal resolution did improve the Gulf Stream separation (see Chassignet and Marshall (2008) and Chassignet and Xu (2017) for a review) in all models, but not necessarily the representation of the Northwest corner circulation (Figure 16). FSU-HYCOM is the only model
that has a good representation of the Northwest corner in both the low- and high-resolution experiments. Since all the models use the same atmospheric forcing dataset, the difference is solely due to the numerical and physical choices made by each modeling group.

As expected, there is a significant increase in the SSH variance as resolution is increased and the eddying solution SSH variance maps are much closer to the observations (top panel) than their low-resolution counterparts. In the high-resolution
experiments (~0.1°), the variability is however still lower than observed, especially in the gyre interiors and in the experiments that use relative winds (NCAR-POP, AWI-FESOM, and IAP-LICOM). This underestimation is thus partly a consequence of the well-known "eddy-killing" effect which results from taking into account the shear between atmospheric wind and ocean current when computing the wind stress and which can reduce the kinetic energy by as much as 30% (see discussion in section 2a). However, the use of absolute wind in FSU-HYCOM is not sufficient to raise the level of surface variability to that of the



observations and Chassignet and Xu (2017) argues that one actually needs to significantly increase the resolution (~0.01°) in order to resolve the submesoscale instabilities that can energize the mesoscale (Callies et al., 2016) and therefore enhance eddy kinetic energy comparable to the mesoscale AVISO observations. It is more physical to take into account the vertical shear between atmospheric winds and ocean currents when computing the wind stress (see Renault et al., 2019, for a review) as it allows for a better representation of western boundary current systems (Ma et al., 2016), especially the Agulhas Current

retroflection and associated eddies (Renault et al., 2017). In FSU-HYCOM, the Agulhas eddies are too regular and follow the same pathway. The use of relative winds not only increases the eddy decay, it also impacts the location of the Agulhas retroflection and where the eddies are formed.

**Figure 16: Top panel: Mean 1993-2018 SSH and variance AVISO. Middle panel: Difference between the mean modeled SSH and**
**AVISO SSH. Bottom panel: Modeled variance derived from 5-day average outputs. The low-resolution IAP-LICOM SSH variance**
**was not provided.**





The eddy kinetic energy maps for the high-resolution simulations are displayed in Figures 17-20 for the surface, at 700
m, and at 1000 m. Observed surface EKE maps are either derived from altimetry (Figure 17a) or drifters (Figure 17b). Because
of the irregular sampling, this necessarily implies some type of smoothing in space and time. In the case of the AVISO along-
track altimetry observations, they are optimally interpolated on a 0.25° grid which filters scales less than 150 km (due to track
separation and measurement noise and errors) and time scales less than 10 days (repeat cycle of the altimeters). There is a
significant reduction in EKE when computing it using 5- or 10-day average outputs (as opposed to online at every time step)
as shown in Figure 18 for FSU-HYCOM and NCAR-POP. In addition to a significant EKE reduction in the most active regions,
the time averaging removes much of the small-scale variability associated with inertial motions and ageostrophic effects. There
is not a big difference between the 5-day and the 10-day maps and for the rest of this section, we will use 5-day average model
outputs when comparing the EKE to observations. Overall, as for the SSH variability, the EKE is larger in FSU-HYCOM
because absolute winds are used to force the models. As already discussed, the use of relative winds does improve the pathway
for the Agulhas current eddies, but from Figure 17, there is also a significant reduction of EKE in the ACC, and it also
suppresses variability in many areas. This is especially true for the Indian Ocean, in the tropics, and west of the Hawaiian
Islands.

Most model–observations comparisons usually focus on the surface fields because of very sparse spatiotemporal sampling
at depth (see for example the EKE derived at 700 m from several years of SOFAR float measurements in the Gulf Stream
region (Richardson. 1993) or the EKE distribution from ARGO floats at 1000 m (Ollitrault and Colin de Verdière, 2014)).
Scott et al. (2010) evaluated the total kinetic energy simulated in eddying ocean models (on the order of 0.1° horizontal
resolution) relative to moored current records. They found that the models agreed within a factor of 2 above 3500 m depth and
within a factor of 3 below 3500 m depth. Thoppil et al. (2011) show that surface and the abyssal EKE increase with resolution
and clearly that better upper-ocean EKE allows strong eddy-driven abyssal circulation. This also means, that while the EKE
at depth in the high-resolution experiments is a significant improvement over the quasi non-existent EKE of the coarse-
resolution simulations, it is still significantly less than very limited observations (Richardson, 1993; Ollitrault and Colin de
Verdière, 2014). Significantly higher resolution may be necessary in order to obtain a level of EKE close to the observations
(see Chassignet and Xu (2017) for a discussion). When using relative winds at this resolution as in NCAR-POP, AWI-FESOM,
or IPA-LICOM, there is very little EKE at depth when compared to FSU-HYCOM forced with absolute winds (Figures 19
and 20).







**Figure 17: Surface EKE from a) AVISO and b) drifters observations. Surface EKE calculated from 5-day average fields for c) FSU-HYCOM, d) NCAR-POP, e) AWI-FESOM and f) IAP-LICOM.**

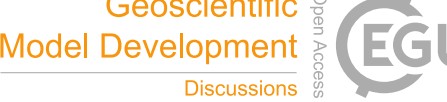

**Figure 18: Comparison of surface EKE calculated from integrated kinetic energy, 5-day average fields and 10-day average fields in FSU-HYCOM and NCAR-POP.**



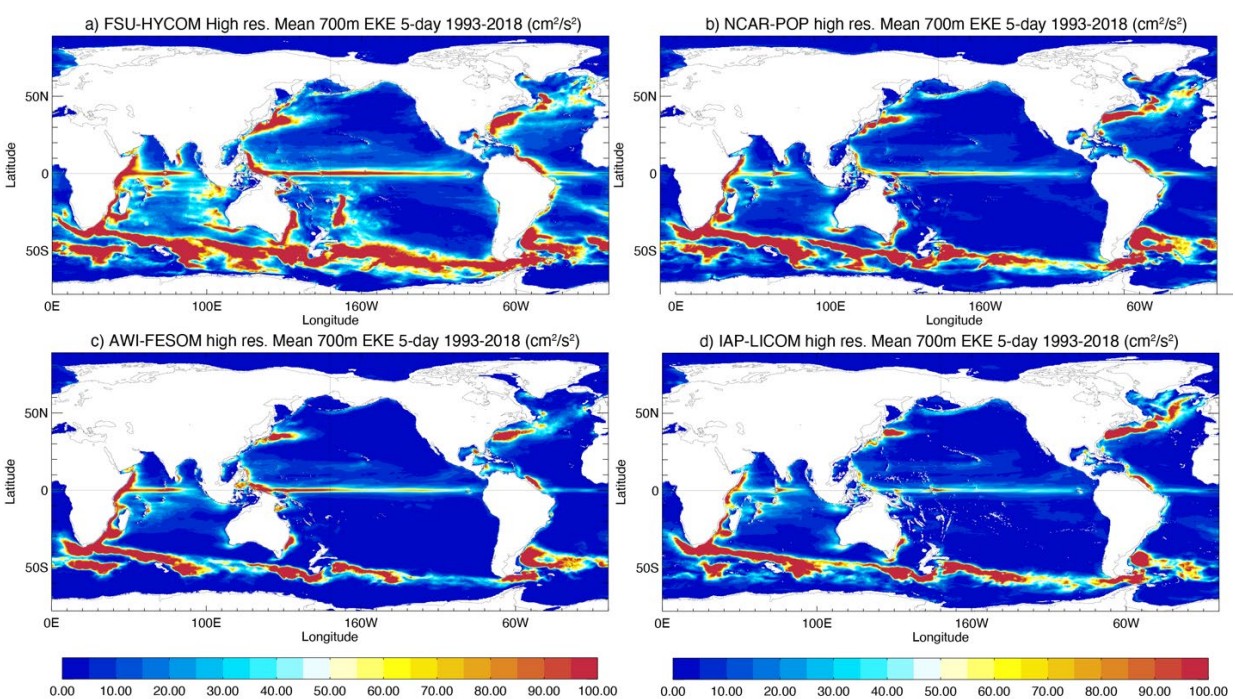

**Figure 19: EKE at 700 m calculated from 5-day average fields in a) FSU-HYCOM, b) NCAR-POP, c) AWI-FESOM and d) IAP-LICOM.**

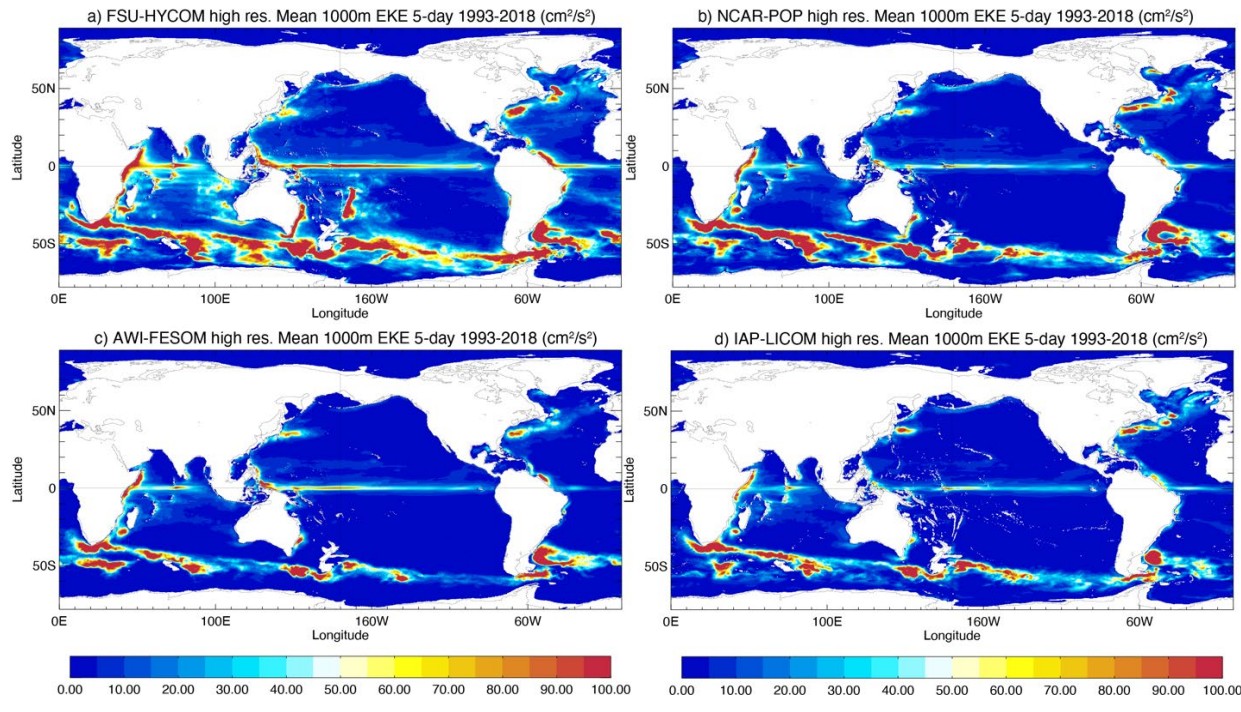


**Figure 20: EKE at 1000 m calculated from 5-day average fields in a) FSU-HYCOM, b) NCAR-POP, c) AWI-FESOM and d) IAP-LICOM.**





### 4.2 Sea Surface Temperature (SST) and Sea Surface Salinity (SSS) deseasonalized variance

Surface temperature and salinity show sensitivities to resolution that are largely consistent with those described above for SSH in terms of both mean state and variability. All models show greatly enhanced SST variance (computed from deseasonalized monthly values spanning 1980-2018) at high resolution in regions of high mesoscale eddy activity, including: the Southern Ocean, the Agulhas Current retroflection, the Brazil-Malvinas Confluence region, the Kuroshio and its extension across the North Pacific, and the Gulf Stream and its NAC extension (Figure 21). A global, satellite-based SST dataset spanning

1982-2018 provides some measure of ground truth for comparison, but the sampling limitations of microwave and infrared measurements of SST restrict the observed estimate to a (¼°) grid that is 2-3 times coarser than the high-resolution model grids considered here (OISST.v2; Reynolds et al., 2007; Banzon et al., 2014).

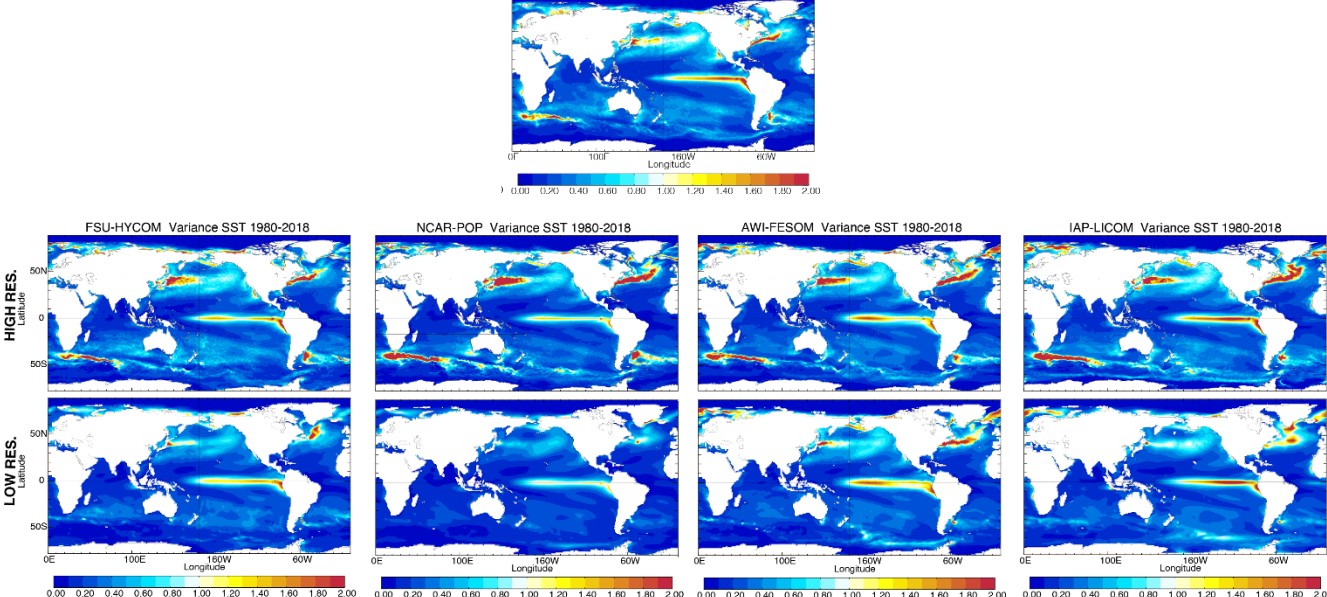

**Figure 21: Modeled and observed deseasonalized SST variance.**

    As was noted for SSH, the high-resolution FSU-HYCOM simulation generally exhibits the highest SST variance in the midlatitude gyre interiors compared to other models, but it has lower SST variance in strongly eddying regions such as the Agulhas retroflection, Kuroshio, and NAC. Overall, the high-resolution FSU-HYCOM shows the closest match to the observed benchmark. The improved structure of SST variance in the subpolar Atlantic in FSU-HYCOM is related to the improved NAC

pathway in that model (discussed above). Compared to OISST.v2, the low-resolution models generally underestimate SST variance while the high-resolution models tend to overestimate the variance, particularly in the western boundary currents, Southern Ocean, and Agulhas retroflection region. The resolution sensitivity is most dramatic in the NCAR-POP model, whose low-resolution version shows the weakest SST variance of any of the simulations considered here. The overestimation of SST





variance has been noted before in eddy-resolving coupled simulations (e.g., Small et al. 2014), but it is not clear whether this
discrepancy is attributable to deficiencies in the models or in the observation-based estimates which cannot fully resolve ocean
mesoscale variability. The absence of a dynamic (and ocean mesoscale-resolving) atmosphere in these simulations may
partially explain the overly high SST variance insofar as important eddy-damping processes (Ma et al., 2016) are missing in
this experimental framework. It is interesting that all high resolution simulations generate more SST variance in the Kuroshio
region than seen in OISST.v2, and yet they all show less variance than observed in the northeast Pacific and along the
southeastern branch of the North Pacific gyre (near Hawaii). There is an indication of slightly enhanced ENSO-related tropical
Pacific SST variance when going from low to high resolution, but there is not a systematic change in the spatial structure of
this variance which appears to depend mainly on model formulation. The representation of high extratropical SST variance
along the eastern boundaries of each of the major ocean basins shows robust improvement with resolution across the different
models. The high resolution versions all show higher (and more realistic) variance in the following locations: the Benguela
Current region in the eastern South Atlantic; the coastal region off the west coast of North America and Baja; the Leeuwin
Current region off the west coast of Australia; and the Canary Current region off the west coast of North Africa. Improvements
in simulated SST variability in these regions is more apparent than improvements in the mean state (Figure 9).  The overall
picture of greatly enhanced SST variance at eddying resolution is an important, but not unexpected, result that has significant
implications for climate modeling because resolving air-sea interactions at the ocean mesoscale has been shown to result in
qualitatively different coupled model behavior (e.g., Bryan et al., 2010; Ma et al., 2016, 2017).

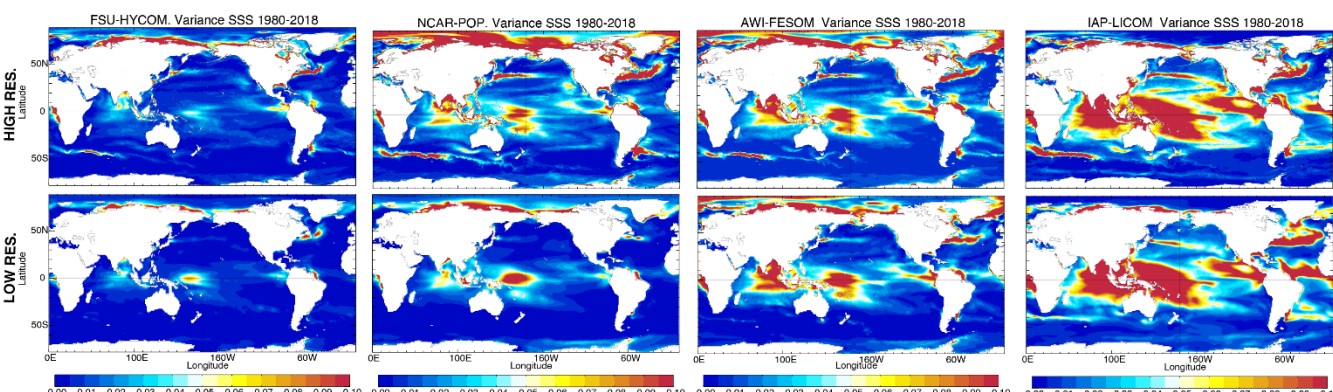

**Figure 22:  Modeled and observed deseasonalized SSS variance.**

While surface salinity generally exhibits globally enhanced variance at high resolution, particularly in the eddy-rich
locations already highlighted above, the SSS variance plots are striking in that they show considerably more sensitivity to
model formulation than to model horizontal resolution (Figure 22). The increase in SSS variance at both resolutions across
model systems—from FSU-HYCOM, to NCAR-POP, to AWI-FESOM, and to IAP-LICOM—is likely related to the steady
increase in surface salinity restoring timescale (Table 1). In the absence of reliable long-term global SSS observations, it is





difficult to say which model formulation is best or whether high resolution improves the simulation of surface salinity variability.

**4.3 Upper ocean heat and salinity content mean biases and deseasonalized variance**

The temperature and salinity distributions in the upper 700 m are less constrained to observations than are the SST (which is restored to time-varying observed values through the sensible heat flux) and the SSS (which is restored to observed climatology through the artificial salinity restoring flux). Therefore, vertically-averaged upper ocean heat content (UOHC) and salt content (UOSC) biases tend to be larger than surface biases, particularly at low latitudes (cf. Figures 9, 23, and 24). The

UK Met Office EN4 ocean reanalysis (Good et al., 2013) provides an observational benchmark for UOHC and UOSC mean and variance, but again considerable caution is warranted when interpreting model-observation discrepancies. The spatial resolution of this analysis is only 1º, and it relaxes to climatology in data-sparse regions, which are extensive in the pre-Argo era (prior to 2000).

Both FSU-HYCOM and NCAR-POP show significant and near ubiquitous mean bias reduction for UOHC and UOSC

when resolution is increased (Figures 23 and 24). In contrast, AWI-FESOM and IAP-LICOM show mixed results, with bias reduction in some regions (e.g., the Southern Ocean) offset by bias increase elsewhere. Regions of high observed UOHC variance include all the regions of high observed SST variance mentioned above as well as some variance hotspots in the subtropical western Pacific, the Tasman Sea, the subtropical south Indian Ocean, and the western tropical Atlantic (Figure 23). The high-resolution models tend to show improved representation of these subsurface variance hotspots as well as enhanced

variance in high EKE regions (western boundary currents, Agulhas, Southern Ocean, etc.). The high resolution IAP-LICOM model stands out for its globally high (perhaps unrealistically high) UOHC variance. This upper ocean variance bias in IAP-LICOM is even more evident in UOSC (Figure 24), although the limited sampling of 0-700m salinity in the EN4 product implies that the observed variance estimate is almost certainly a gross underestimate. There is a hint (especially apparent in NCAR-POP and IAP-LICOM) of higher resolution resulting in enhanced salinity variance along subtropical cell spiciness

pathways from the eastern extratropical Pacific to the tropical western Pacific (e.g., Yeager and Large, 2004).



**Figure 23: Top panel: Mean 1980-2018 0-700 m temperature and variance from EN4. Middle panel: Difference between the mean modeled 1980-2018 0-700 m temperature and EN4. Bottom panel: Deseasonalized 0-700 m temperature modeled variance.**





**Figure 24: Top panel: Mean 1980-2018 0-700 m salinity and variance from EN4. Middle panel: Difference between the mean**
**modeled 1980-2018 0-700 m salinity and EN4. Bottom panel: Deseasonalized 0-700 m salinity modeled variance.**

### 4.4 Zonal AMOC mean and variance

Although the magnitude and the temporal evolution of the AMOC transport, as shown in Figures 14a-b, differ significantly

between different models and different horizontal resolutions, its time mean spatial structure and variance are quite similar

among all the simulations as shown in Figure 25 for the 1980 to 2018 period. All simulations show a positive upper cell and a

negative lower cell. In the low-resolution simulations, the upper cell exhibits a maximum transport near 1000 m and 35°-40°N.

The lower cell has its maximum transport near 3500-4000 m and is much weaker overall (~2 Sv), except for IAP-LICOM (~6

Sv in the subtropical North Atlantic). In high-resolution simulations, the upper cell extends deeper in all 4 models, which is

more consistent with the observations, and the lower cell has a similar magnitude (2-4 Sv). This can be seen more clearly in

Figure 26, which compares the model AMOC profiles for 2004-2018 to the RAPID results near 26.5°N. South of about 20°N,





the high-resolution AWI-FESOM exhibits a weak positive cell near the bottom that is unrealistic and is different from other

models.

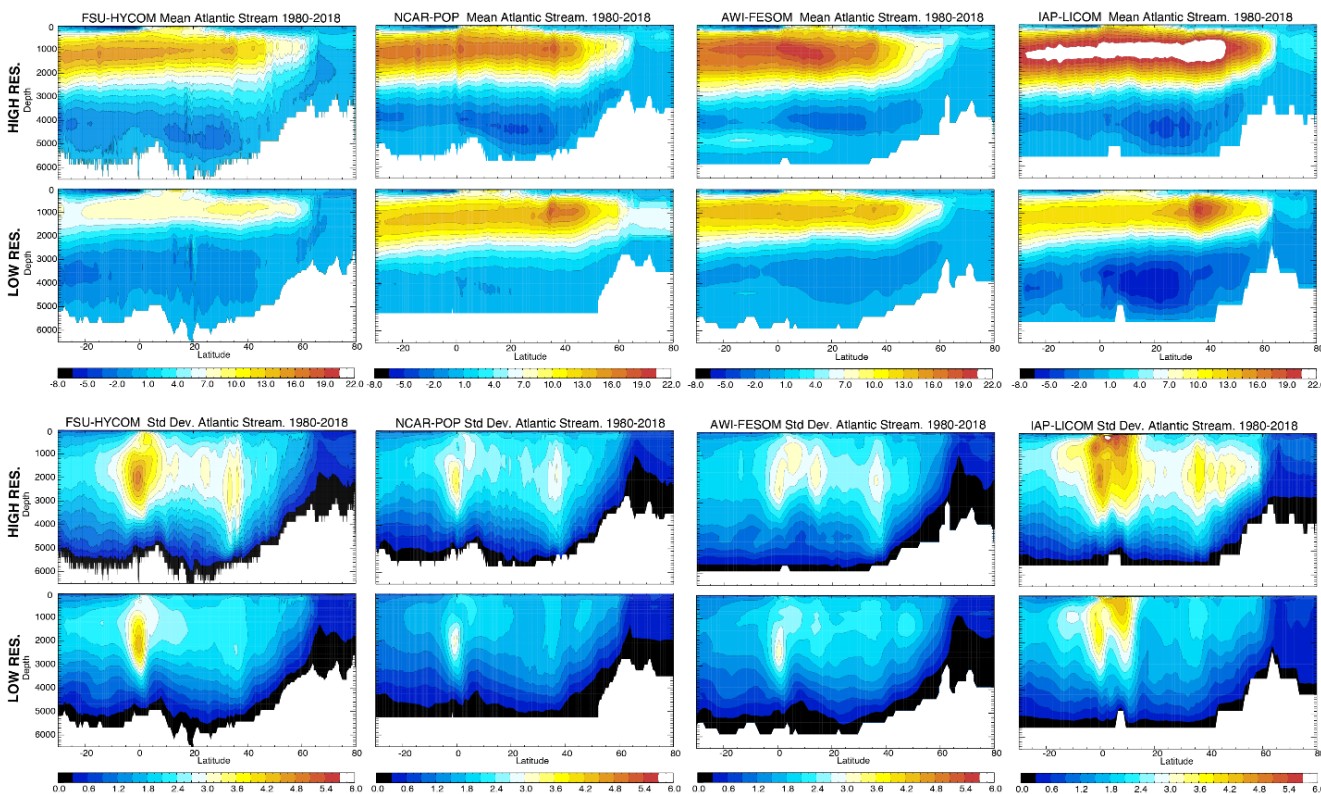

**Figure 25: (Top panels) Modeled time mean meridional overturning streamfunction $\psi_z(y, z)$ over 1980-2018 in four models with high and low resolutions. (Lower panels) Modeled standard deviation of the overturning streamfunction $\psi_z(y, z)$ over 1980-2018**
**based on monthly mean results.**

The AMOC standard deviation shows a similar pattern between different resolutions and different models (Figure 25). The variability is overall highest near the Equator (IAP-LICOM has the strongest variability near 10°N that differs from the other three). This is due to the large seasonal variability of the AMOC, associated with the migration of the Intertropical Convergence Zone (ITCZ) and the changes in wind patterns and Ekman transport (Xu et al., 2014). The standard deviations

based on annual means does not show such a maximum near the Equator (see Figure 7 in Hirschi et al. 2019 for results based on different averaging scales). The variability is much weaker beyond the equatorial region. In the low-resolution simulations, the standard deviation is typically 2 Sv and has a slight elevation near 40°N. In the high-resolution simulations, the standard deviation is about 3 Sv and clearly shows a secondary maximum near 40°N. The difference between the low- and high-resolution simulations near the 40°N highlights the impact of the meandering and mesoscale eddy variability of the Gulf Stream

to AMOC variability.

The AMOC dominates the meridional heat transport in the Atlantic Ocean (e.g., Msadek et al., 2013; Xu et al., 2016). Thus, the high-resolution simulations have a higher heat transport in the Atlantic than the low-resolution counterparts (Figure



27). The maximum northward heat transport in 20-30°N is about 1.0 PW in three high-resolution models, 0.6-0.8 PW in low-resolution models, and 1.25-1.3 PW in observations (Johns et al., 2011; Trenberth et al., 2019). In other basins such as the

Indo-Pacific and the Southern Oceans, higher resolution does not necessarily lead to higher heat transport (Figure 26a-c). But in general, the meridional heat transport in high-resolution is closer to observations than the low-resolution simulations (Figure 27).

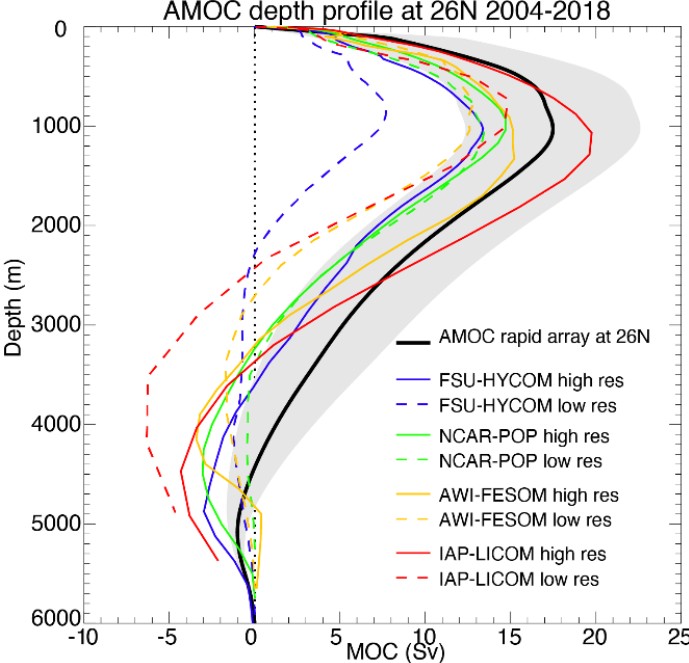

**Figure 26:  AMOC profile at 26°N.**

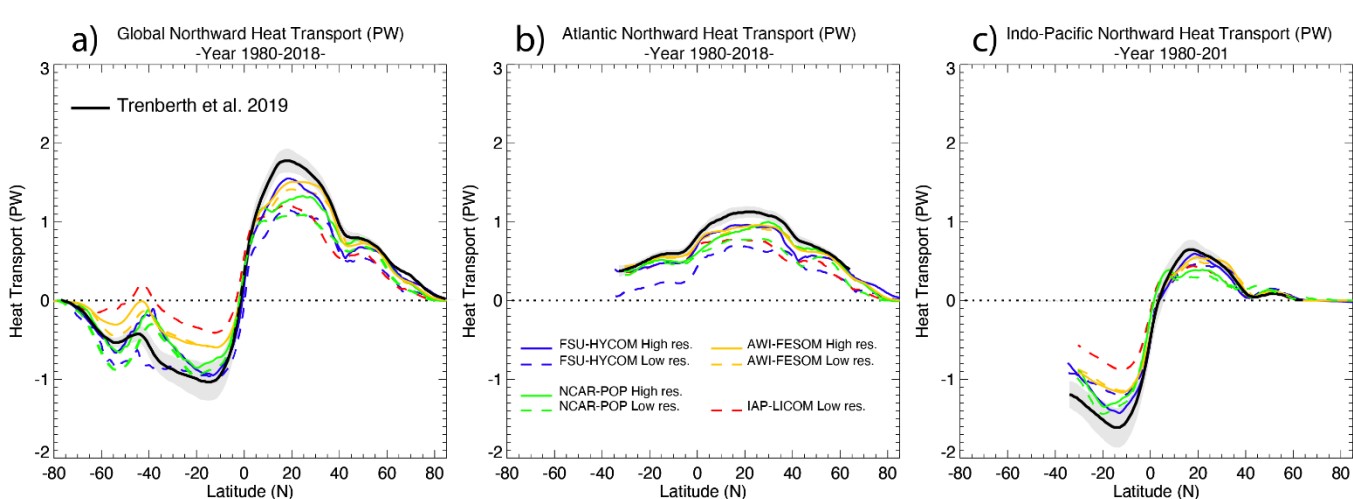


**Figure 27:   Modeled time mean meridional Heat Transport in PW over 1980-2018 for a) global ocean; b) Atlantic Ocean; and c) Indo-Pacific Ocean. Black lines are the latest observational results from Trenberth et al. (2019) with uncertainties.**



## 4.5 Northern and southern hemisphere sea-ice and variance

The observed spatial pattern of winter sea-ice concentration (averaged from 1980 to 2018 for March) in the northern
hemisphere is well simulated in all the models with both low- and high-resolutions, although there tends to be greater
consistency among the high-resolution models. The simulated winter sea-ice spatial distribution is very similar between the
models and setups with different resolutions. There is only very minor difference in the exact location of sea-ice outer edges
in the sub-Arctic seas (Figures 28 and 29). Wintertime agreement is particularly good, with only low-resolution FSU-HYCOM
deviating from the other models. In summer, all the models can reproduce the observed spatial pattern with high sea-ice
concentration north of Greenland and low concentration towards the Arctic marginal seas. However, the difference between
the models is more significant than in winter, especially in the shelf regions, which is true in both the low- and high-resolution
experiments. The sea-ice concentration in summer also does not change much with changing resolution, except for IAP-
LICOM, which has higher concentration in most parts of the Arctic Ocean in its high-resolution setup.

In both seasons, the northern hemisphere sea-ice thickness is the largest north of the Greenland and decreases towards the
Eurasian coast. This pattern is reproduced in the models for both resolutions, except for the high-resolution setup of IAP-
LICOM, which has its thickest ice centered around the North Pole (Figures 28 and 29). The sea-ice thickness is underestimated
for both seasons in FSU-HYCOM and NCAR-POP, especially in summer, while it is overestimated for the winter season in
the East Siberian Sea sector in AWI-FESOM. There is some sensitivity to model resolution for the simulated sea-ice thickness,
but the difference due to changes in the resolution is smaller than the difference between the model results and the observation.

As the sea-ice concentration is close to 100% in winter inside the Arctic Ocean, its variance (Figures S1-S8) is very small
in this season, as consistently shown by all the model simulations. In summer, the variance of sea-ice concentration is high
along the outer sea-ice edge. As the summer sea-ice edges in FSU-HYCOM and NCAR-POP are farther from the coast than
in AWI-FESOM, the bands of high variance are located at larger distance from the coast in the former two models. The
variance does not change much with changing resolution in the models, except for IAP-LICOM, because its sea-ice
concentration is quite different when different resolutions are used. The variance of northern hemisphere sea-ice thickness is
stronger along the coastlines in the other three models. Along the Canadian coastline, the thickness variance is relatively high
for both seasons in these models. In winter, the thickness variance is also high along the East Siberian Sea coast in FSU-
HYCOM and NCAR-POP. In summer, the thickness variance is high over the Eurasian continental shelves in AWI-FESOM,
but not in the other models, because these models have too thin summer ice compared to the observation in these regions.
Similar to the finding for the mean sea-ice concentration and thickness, their variance is not very sensitive to the model
resolution and the difference between models is more pronounced (Figures S1-S8).

In the southern hemisphere (Figures 30 and 31), the observed spatial pattern of sea-ice concentration in winter is well
represented by the models. Only FSU-HYCOM has large difference in winter sea-ice concentration in the southern hemisphere
when changing the model resolution. The concentration is too low compared to the observation in its low-resolution setup, and
it is more realistic when the resolution is increased. In summer, the sea ice concentration in the southern hemisphere is not





well reproduced in most models, except for AWI-FESOM. The high summer sea-ice concentration in the Weddell and Ross Seas is only reasonably simulated in AWI-FESOM. The high-resolution setup of IAP-LICOM has too high concentration in a too wide band. The high variability of Antarctic sea-ice by region that obscures the overall Antarctic sea ice change is apparent in the high variance of the models in summertime Antarctic, but there is little agreement about pattern. The low-resolution,

high-resolution model pairs tend to agree more than the ensemble of low-resolution or ensemble of high-resolution models, indicating that sea ice model formulation differences may exceed the effects of resolution in these patterns.

Sea ice is thicker along the Antarctic coast and thinner at its outer edge at the north. The models can reasonably capture this spatial pattern (Figures 30 and 31). In winter, the sea-ice is too thin in the low-resolution setup of FSU-HYCOM, which is improved in its high-resolution setup. NCAR-POP and AWI-FESOM has much less sensitivity to model resolution for the

Antarctic sea ice thickness than in the other models. In summer, sea-ice thickness in the Southern Ocean is significantly underestimated in most areas by the models. It is closer to the observation in AWI-FESOM, although it is thinner than the observation in most regions. The sea-ice thickness in the high-resolution setup of IAP-LICOM is significantly overestimated in both seasons, which is also manifested in the time series of sea ice volume (Figure 13).

The variance (Figures S1-S8) of winter sea ice concentration in the southern hemisphere shows a circumpolar band of

high values, located along the outer sea ice edge. In FSU-HYCOM, there is high variance in the southeastern Weddell Sea, where the mean winter sea ice concentration is too low. This implies formation of large polynya in this region in some years. This feature is eliminated in the high-resolution FSU-HYCOM setup. In summer, the sea-ice concentration variance is quite different between the models. In particular, the variance is very low in the low-resolution FSU-HYCOM setup because it has much less sea ice than in other simulations. The variance of winter sea-ice thickness in the southern hemisphere is the most

pronounced along the sea ice outer edge and the eastern Antarctic Peninsula coast, and in the western Antarctic sector. The models are consistent in the location of large variance, although there are differences in the strength of the variance. In summer, strong variance is present mainly in the Weddell Sea and along the coast of the western Antarctic. The models do show some sensitivity to the model resolution for variance of the sea-ice concentration and thickness in the southern hemisphere, especially in FSU-HYCOM. However, the difference between models is even larger. As IAP-LICOM significantly overestimated the

sea-ice thickness in its high-resolution setup, the variance is also much higher than in other models (Figures S1-S8).



**Figure 28:  Northern hemisphere winter mean ice cover and thickness.**





**Figure 29: Northern hemisphere summer mean ice cover and thickness.**




**Figure 30: Southern hemisphere summer mean ice cover and thickness.**




**Figure 31: Southern hemisphere winter mean ice cover and thickness.**



## 5 Summary and Discussion

The overall goal of this paper is to assess the robustness of climate-relevant improvements in ocean simulations (mean and variability) associated with moving from coarse (~1º) to eddy-resolving (~0.1º) horizontal resolutions. It also lays out a set of basic large-scale diagnostics for assessing the relative quality, variability, and sensitivity of high-resolution versus low-resolution ocean and sea-ice models. The emphasis is on the key metrics used in climate modelling – SST, OHC, sea level, salinity, sea ice extent and volume, and circulations that tend to have global impacts (MOC, ACC, ITF) on the modeled climate.

Here these metrics are assessed in a suite of 4 pairs of low-resolution, high-resolution ocean and sea-ice models forced with the latest JRA55-do dataset (Tsujino et al., 2019). These results will provide a useful baseline for future process-focused analyses as well as ocean model development activities at diverse resolutions.

On the whole, the biases in the low-resolution variants are familiar, and are similar in these models to those found when assessing sensitivity to forcing products (Tsujino et al., 2020). Gross features of the bias patterns in low resolution models--

position, strength, and variability of western boundary currents, equatorial currents, and ACC – are significantly improved in the high-resolution models. However, despite the fact that the high-resolution models "resolve" these features, the improvements in temperature or salinity are inconsistent among the different model families and some regions show increased bias over their low-resolution counterparts. SSH variability and near-surface EKE are significantly – even qualitatively – improved in all high-resolution models over their low-resolution counterparts, although all of these models still underpredict

the observed SSH variability and EKE, particularly in the ocean interior, which indicates a need for further refinements in resolution (Chassignet and Xu, 2017), and improvements in less dissipative subgrid schemes for high-resolution models (Pearson et al., 2017) are needed. The results in coupled models in the HighResMIP ensemble (Haarsma et al., 2016) show similar improvements in SSH and SST variability and EKE. Considerable differences in the high-resolution models used here were associated with the use of relative winds versus absolute winds. Renault et al. (2020) show that using relative winds

contributes a feedback that tends to reduce EKE, and the models here are consistent with a 30% or so reduction of EKE when relative winds are used.

One interesting aspect of the high-resolution models versus the low-resolution models was that the interannual variability in the ACC and ITF transport and AMOC was more consistent among the high-resolution models than among the low-resolution models. The ITF transport was especially in agreement among the high-resolution models, indicating that better

representation of the passageways through the Indonesian archipelago is critical. Consistency in all of these transports potentially indicates that higher-resolution models are needed to represent process variability, which may explain some of the past difficulties in comparing the magnitude of these phenomena across coarse-resolution models. However, the mean ACC transport and MOC strength were not in greater agreement among the high-resolution than the low-resolution models, which means that more work remains in evaluating sensitivity to numerics and subgrid-scale schemes for high-resolution models

remains. Furthermore, Danabasoglu et al. (2016) note that low-resolution models come into greater agreement in AMOC variability after more cycles of the CORE forcing – this comparison was limited by the cost of the high-resolution models to





only a single cycling of the forcing. It is unclear if the high-resolution models are in greater agreement only in the first cycle or generally. The short duration of a single forcing cycle limits the comparison of the decadal changes that are emphasized in Danabasoglu et al. (2016), so the improved agreement discussed here among high-resolution models is year-by-year rather
than decade-by-decade. Nonetheless, the high-resolution models had systematically stronger and more variable AMOC, which was in better agreement with observations, both in maximum overturning and profile, than the low-resolution models.

From a climate modeling perspective, ocean heat content, sea level, and sea ice stability are key metrics. There is some indication that the high-resolution models may warm more quickly below 700 m than the low-resolution models, indicating errors in parameterizations of vertical eddy heat transport. Griffies et al. (2016) found similar sensitivity to resolution in a
model hierarchy which they explained as resolved eddies versus parameterized eddies affecting vertical heat transport. However, warming between 0-700 m and global mean sea level rise were not systematically different across model families between the low-resolution and high-resolution models. Regional sea level rise, however, was significantly more variable in the high-resolution models where mesoscale eddies and variability of western boundary and equatorial currents make an impact on regional sea level rise. Thus, it is critical to use high-resolution models to assess the ocean dynamic sea level (Gregory et
al., 2019) contribution to regional sea level rise. In terms of sea ice, high-resolution models were better than or equal to low-resolution models in reproducing observations of extent and volume (except for the IAP-LICOM model), and the high-resolution models were more consistent in reproducing observed trends and regional patterns in sea ice.

Low-resolution versus high-resolution comparisons are often motivated by identifying persistent patterns across multiple models of bias or improvements with resolution. However, as apparent in many metrics, there is no overall consistency among
either the low-resolution or high-resolution ensemble from which to draw simplistic conclusions. This paper therefore does not dwell on the detailed differences among these models' numerics and parameterizations, but is instead intended to serve as a benchmark for future studies comparing and improving different schemes in any of these models or similar ones. The numerics and parameterizations of these models are in continual development in both low-resolution and high-resolution versions. Other modeling centers have expressed interest in participating in this protocol in the future, and this paper provides
a comparison basis. The set of tests performed here are a benchmark against which development changes can be assessed.

The models used in the present study are not coupled to active atmospheres, biospheres, land models, nor land ice models. Thus, many important feedbacks and aspects of climate modeling are not addressed here. However, these same feedbacks complicate diagnosis of the source of model biases and resolution sensitivity, which is where this study intends to contribute.

**Code and data availability**

The forcing dataset for OMIP-2 is available through input4MIPs (https://esgf-node.llnl.gov/search/input4mips/) – see Table 1 in the supplement of a list of the files. An archive of the model outputs and the scripts used to process data and generate figures are available at https://doi.org/10.5281/zenodo.3685918.



**Author contribution**

EC, SY, and BFK proposed and led this evaluation study. AB processed the model outputs and produced the figures. The

following authors are responsible for the individual models, simulations, and diagnostics: AB, XX, and EPC for FSU-HYCOM; SY, FC, WK, and GD for NCAR-POP; QW, SD, NK, and DS for AWI-FESOM; HL, YL, and PL for IAP-LICOM3. All authors contributed to the writing and editing processes.

**Competing interests**

The authors declare that they have no conflict of interest.

**Acknowledgements**

The FSU contribution was supported by the National Oceanic and Atmospheric Administration (NOAA) Earth System Prediction Capability Project (Award NA15OAR4320064), NOAA Climate Program Office (CPO) MAPP Program (Award 5NA15OAR4310088), and NSF Physical Oceanography Program (Award 1537136). The NCAR contribution was supported by the NOAA CPO Climate Variability and Predictability (CVP) Program. NCAR is a major facility sponsored by the US

National Science Foundation (NSF) under Cooperative Agreement No. 1852977. The AWI contribution was supported by the projects S1 (Diagnosis and Metrics in Climate Models) and S2 (Improved parameterizations and numerics in climate models) of the Collaborative Research Centre TRR 181 "Energy Transfer in Atmosphere and Ocean" funded by the Deutsche Forschungsgemeinschaft (DFG, German Research Foundation) – project no. 274762653, Helmholtz Climate Initiative REKLIM (Regional Climate Change) and European Union's Horizon 2020 Research & Innovation programme  through grant

agreement No. 727862 APPLICATE. The IAP contribution was supported by the National Natural Science Foundation of China (Grants No. 41931183 and 41976026). D. Sein was supported by the state assignment of FASO Russia (theme 0149-2019-0015).

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



**Table 1: Model parameters for the low-and high-resolution configurations**

| | *Horizontal grid spacing* | *Explicit horizontal viscosity* | *Explicit vertical viscosity* | *Explicit horizontal diffusivity* |
|---|---|---|---|---|
| *FSU-HYCOM low-resolution* | 0.72° | A= max (Smagorinsky, Laplacian $A_2$) + biharmonic $A_4$ with Smagorinsky = 0.1 $\Delta x^2$ × deformation tensor, $A_2$=0.03$\Delta x$ m²/s, and $A_4$= -0.05 $\Delta x^3$ m⁴/s | Background viscosity of 3 $10^{-5}$ m²/s² in KPP | Laplacian (0.03$\Delta x$ m²/s²) |
| *FSU-HYCOM high-resolution* | 1/12° (8 km at the Equator, 6 km at mid-latitudes) | A= max (Smagorinsky, Laplacian $A_2$) + Biharmonic $A_4$ with Smagorinsky = 0.05 $\Delta x^2$ × deformation tensor, $A_2$=20 m²/s, and $A_4$=-0.01$\Delta x^3$ m⁴/s. | Background viscosity of 3 $10^{-5}$ m²/s² in KPP | Laplacian (0.005$\Delta x$ m²/s²) |
| *NCAR-POP low-resolution* | Nominal 1° bipolar grid with tropical refinement down to 1/4° | Anisotropic horizontal viscosity (see Danabasoglu et al., 2012, for details). | Spatially variable background viscosities between $10^{-5}$ and 3 $10^{-4}$ m²/s occurring at the equator and at about 30˚ of latitude. | Isopycnal diffusivity with enhanced values that can be as large as 3000 m²/s in the upper ocean and diminish to 300 m²/s by a depth of about 2000 m. |
| *NCAR-POP high-resolution* | 1/10° tripolar grid | Horizontal biharmonic (-2.7 $10^{10}$ m⁴/s) | Background viscosity of $10^{-4}$ m²/s at the surface up to $10^{-3}$ m²/s in the abyss in KPP. | Horizontal biharmonic diffusion (-3$10^9$ m⁴/s) |
| *AWI-FESOM low-resolution* | Nominal 1°; refined at the equator and around Antarctica; 25 km north of 45°N (see Figure 1a) | Biharmonic Smagorinsky = - $\Delta x^4$ × deformation tensor/32 | Background viscosity of $10^{-4}$ m²/s² in KPP | 0 |
| *AWI-FESOM high-resolution* | Scaled with the observed sea surface height variance, ranging from about 10 to 50 km (see Figure 1b) | Biharmonic Smagorinsky = - $\Delta x^4$ × deformation tensor/32 | Background viscosity of $10^{-4}$ m²/s² in KPP | 0 |
| *IAP-LICOM low-resolution* | 1° (110 km in longitude, about 110 km at equator and 80 km at mid-latitudes) | Laplacian (5400 m²/s) | Background viscosity of 2 $10^{-6}$ m²/s as in Canuto et al. (2001, 2002) with an upper limit of 2 $10^{-2}$ m²/s | Laplacian (5400 m²/s) |
| *IAP-LICOM high-resolution* | 1/10° (11 km in longitude, about 11km at equator and 8 km at mid-latitudes) | Biharmonic (-2.8 $10^{10}$ m⁴/s) | Background viscosity of 2 $10^{-6}$ m²/s as in Canuto et al. (2001, 2002) with an upper limit of 2 $10^{-2}$ m²/s | Biharmonic (-2.8 $10^{10}$ m²/s) |



**Table 1: Model parameters for the low-and high-resolution configurations (continued)**

|  | *Isopycnal scheme, e.g. GM* | *Mixed layer scheme* | *Momentum advection scheme* | *Tracer advection scheme* | *Time stepping scheme* |
|---|---|---|---|---|---|
| *FSU-HYCOM low-resolution* | Laplacian ($0.01\Delta x$ m$^2$/s) + biharmonic (-$0.02\Delta x^3$ m$^4$/s) thickness diffusion | KPP | 2$^{nd}$ order FCT | 2$^{nd}$ order FCT | Split-explicit leapfrog with Asselin filter (0.125) |
| *FSU-HYCOM high-resolution* | Biharmonic thickness diffusion (-0.015 $\Delta x^3$ m$^4$/s) | KPP | 2$^{nd}$ order FCT | 2$^{nd}$ order FCT | Split-explicit leapfrog with Asselin filter (0.125) |
| *NCAR-POP low-resolution* | GM + submesoscale parameterization | KPP | 2$^{nd}$ order centered | 2$^{nd}$ order centered | 2$^{nd}$ order leapfrog scheme with Asselin filter |
| *NCAR-POP high-resolution* | None | KPP | 2$^{nd}$ order centered | 3$^{rd}$ order upwind | 2$^{nd}$ order leapfrog scheme with averaging time step |
| *AWI-FESOM low-resolution* | Laplacian Redi and thickness diffusion, diffusivity flow-dependent in the range of 0 to 1500 m$^2$ as implemented in Danabasoglu et al. (2008). | KPP | Taylor-Galerkin | 2$^{nd}$ order FCT | Pressure split; implicit SSH |
| *AWI-FESOM high-resolution* | Scaled with the observed sea surface height variance, ranging from about 10 to 50 km (see Figure 1b) | KPP | Taylor-Galerkin | 2$^{nd}$ order FCT | Pressure split; implicit SSH |
| *IAP-LICOM low-resolution* | Both Redi and GM with coefficient computed as in Ferreira et al. (2005) | Canuto et al. (2001, 2002) | Two step preserving shape (Yu, 1994) | Two step preserving shape (Yu, 1994) | Split-explicit leapfrog with Asselin filter (0.2 for barotropic; 0.43 for baroclinic; 0.43 for tracer) |
| *IAP-LICOM high-resolution* | 1/10° (11 km in longitude, about 11km at equator and 8 km at mid-latitudes) | Canuto et al. (2001, 2002) | Two step preserving shape (Yu, 1994) | Two step preserving shape (Yu, 1994) | Split-explicit leapfrog with Asselin filter (0.2 for barotropic; 0.43 for baroclinic; 0.43 for tracer) |





1315 **Table 1: Model parameters for the low-and high-resolution configurations (continued)**

| | *Bottom drag* | *Surface wind-stress* | *Vertical coordinates* | SSS restoring |
|---|---|---|---|---|
| *FSU-HYCOM low-resolution* | 0.72° | Relative wind stress | 41 hybrid layers | 30 m/60 days to monthly GDEM |
| *FSU-HYCOM high-resolution* | Quadratic bottom drag $C_b(|U| + U_{bar})\vec{U}$ with $C_b$=1.5 $10^{-3}$ and $U_{bar}$=0.05 m/s | Relative wind stress | 36 hybrid layers | 30 m/60 days to monthly GDEM |
| *NCAR-POP low-resolution* | Quadratic bottom drag with $C_b = 10^{-3}$ | Relative wind stress | 60 z-levels | 50 m/1 year to monthly WOA13 |
| *NCAR-POP high-resolution* | Quadratic bottom drag with $C_b = 10^{-3}$ | Relative wind stress | 62 z-levels with partial bottom cell | 50 m/1 year to monthly WOA13 |
| *AWI-FESOM low-resolution* | Quadratic bottom drag with $C_b$=2.5 $10^{-3}$ | Relative wind stress | 46 z-levels | 50 m/300 days to monthly PHC3.0 |
| *AWI-FESOM high-resolution* | Quadratic bottom drag with $C_b$=2.5 $10^{-3}$ | Relative wind stress | 46 z-levels | 50 m/300 days to monthly PHC3.0 |
| *IAP-LICOM low-resolution* | Quadratic bottom drag with $C_b$=2.6 $10^{-3}$ | Relative wind stress | 30 $\eta$ levels | 50 m/4 years to monthly PHC3.0 (50 m/30 days for the sea ice regions) |
| *IAP-LICOM high-resolution* | Quadratic bottom drag with $C_b$=2.6 $10^{-3}$ | Relative wind stress | 55 $\eta$ levels | 50 m/4 years to monthly PHC3.0 (50 m/30 days for the sea ice regions) |