# Peer review of "Impact of horizontal resolution on global ocean-sea-ice model simulations based on the experimental protocols of the Ocean Model Intercomparison Project phase 2 (OMIP-2)"

_Geoscientific Model Development, 2019_

## Referee Comment (RC1) · Anonymous Referee #1 · 20 Apr 2020

This paper shows the impact of horizontal resolution on ocean-sea-ice model simulations by comparing the outputs from four models at low and high horizontal resolutions. Several biases in the low-resolutions models are improved when the resolution becomes high. The impacts of model resolution on temperature and salinity bias depend on the model. Although the model configuration differences between low and high resolutions differ among the four models, several improvements in high-resolution models compared to low-resolution models seem to be robust. On the one hand, the biases of temperature and salinity fields depend on the model and region. The report in this

**GMDD**

paper is informative for readers of GMD. I recommend this paper for publication after some minor modifications.

Comments The time evolution of the global temperature was examined in Section 3.2. Levitus et al. (2005) showed a warming of the world ocean over 1955-2003 based on the observations. The warmings in the models seem to be consistent with the observation including its decadal variations. As well as the sea surface height, the impact of global warming should be discussed.

Figure 9 shows the surface temperature and salinity differences between the simulated fields over 2009-2018 and the initial fields, which discusses about how the simulated field changes from its initial state. Global warming appears to influence much the temperature difference. The model bias compared to the observation should be correctly verified. The simulated surface temperature and salinity are constrained to atmospheric forcing and the restoring to the observation respectively. Because these constrain are not perfect. Therefore, it is worth verifying the surface temperature and salinity biases compared with recent observations such as Argo float in the same period.

Minor comments There are not a few typos in the texts. The lead-author should be responsible to make readable paper. In Section 2, the model descriptions of four models are shown. It seems that different authors responsible to each model wrote the texts for each configuration, which makes the texts unreadable. The configuration of each model should be written in the same writing style.

GMD recommend the unit with negative exponent (e.g. m sˆ-1 instead of m/s).

L82: 'AMOC' appears first. 'Atlantic meridional overturning circulation' should be shown.

L98: $\sigma 2$ should be defined.

L107: $\Delta$x should be defined such as 'zonal grid spacing'.

L145-146: 'm2 s-1' should be modified by using superscript.

L313: 'section 2b,c' should be 'section 3.2 and 3.3'.

Sec.3.4: The surface temperature and salinity averaged over 2009-2018 are compared to the initial state (Fig. 9). On the one hand, the vertical structures averaged over 1980-2018 are compared to the initial state (Fig. 10, 11, 12). Why are the average periods different?

Sec.3.7: Please plot the EKE time series along the ACC, if possible. Were the EKE in the models were increased by the zonal wind intensification since the 1970s.

L580: I cannot find Ollitrault and Colin de Verdière, 2014 in References.

Fig. 27: Please add the curves of heat transport of IAP-LICOM high resolution in Fig. 27.

'summer' and 'winter' in the captions of Fig. 30 and 31 should be 'winter' and 'summer' respectively.

L242, 553, 1170: Renault et al. (2019) should be Renault et al. (2020).

L909-918: Please confirm the sequence of references.

L963 includes two references.

L1204 includes two references.

Table.1: Please show the initial conditions.

Table.1: 'Relative wind stress' should be 'Absolute wind stress' for FSU-HYCOM'.

---

## Referee Comment (RC2) · Anonymous Referee #2 · 20 Apr 2020

General Comments: This is an important paper for the ocean and climate modelling community feeding into wider decisions on the benefits of high resolution ocean components to address climate science questions. The conclusion that the impact of resolution is consistent across model families in terms of circulation but model dependent in terms of temperature and salinity biases is an important one. My main comment is that the high and low resolution versions of the models are not always comparable due to differences in vertical as well as horizontal resolution. It would have been preferable to have changed only the horizontal resolution and associated physics so that the effects

of horizontal resolution can be isolated as stated in the abstract. I would suggest that at very least, this issue should be addressed in the model description and discussion in the context of Stewart et al. (Ocean Modelling, 2017) and included as a future recommendation. I would recommend this paper for publication assuming that this and other specific issues below are addressed.

Specific comments:

L21: See discussion above on isolating the effects of horizontal resolution

L86: Title would be better as Model Descriptions

L93: A summary table of the four models at both resolutions would be useful for the reader

L117: Why is FSU model 41 layers in low res and 36 layers in high res? (see above)

L157: What is the impact in the NCAR model of two additional levels as well as partial step formulation?

L207: I couldn't see any mention in the description of the lack of salt flux normalisation in IAP?

L213: What is the impact of changing from 30 to 55 levels in the high resolution? It seems to me that this is more consistent with resolving baroclinic modes at high resolution (Stewart et al., 2017) but 55 levels is probably insufficient to resolve the first baroclinic mode.

L229: Title would be better as Temporal evolution and model drift

L249: Is there any comment on why the global average KE is so different between NCAR and IAP (which should be fairly comparable in terms of resolution)?

L258: Add this discussion of the salt flux normalization to mode description and specifically in description of IAP

L275: Please correct 'all high resolution models warm faster over the upper 2000m and global temperature than their lower-resolution partners'. This doesn't appear to be true for FSU.

L284: I understand that including this is beyond the scope of this paper, but the authors could recommend and pass to the discussion, that the vertical heat and salt budgets are compared in future as in Griffies et al. (J. Climate, 2015) and Von Storch et al. (Ocean Modelling, 2016)

L335: Is it possible to hypothesise what leads to reductions or increases in the biases and the different model behaviour?

L417: is the similarity of the high resolution model and RAPID AMOC evolutions statistically significant or can it be shown to link with reduced deep convection? Are the convection patterns different across resolutions? I think without further discussion, this point is rather speculative.

L445: Should figure 14c be labelled as AABW cell?

L502: Since the temperature and salinity fields at depth are still evolving at the end of these experiments, it seems more likely that the ACC changes are related to buoyancy drifts. Can you comment on this?

L528: Also comment here about the choice of vertical levels affecting the representation of baroclinic modes (Stewart et al., 2017)

L539: I would argue from figure 16, that there is at least marginal improvement in the representation of the NW corner across the models but it is difficult to see in the multi panle figure

L555: Please comment on Agulhas eddies in other models

L710: Can you comment on the cause of the increased northward heat transport in the Atlantic by looking at the components? Presumably, it is related to a stronger Gulf

[Figure]

Stream transport

L834: Reference should be Griffies et al. (2015) see below. Include here recommendation on diagnosing vertical heat and salt budgets in future experiments?

L848: I think specific comment is merited here on the aim to maintain traceability across resolution hierarchies in terms of numerics and parameterisations as well as vertical resolution

Technical corrections:

L106 (and similar throughout): present as 3 x 10-5

L414: extra 'only' in the sentence

L505: The sentence doesn't make sense as currently written

References:

Griffies, S. M., Winton, M., Anderson, W. G., Benson, R., Delworth, T. L., Dufour, C. O., Dunne, J. P., Goddard, P., Morrison, A. K., Rosati, A., Wittenberg, A. T., Yin, J. J. and Zhang, R., 2015. Impacts on Ocean Heat from Transient Mesoscale Eddies in a Hierarchy of Climate Models, J. Clim., 28, 952-977.

Stewart, K.D., Hogg, A. McC., Griffies, S. M., Heerdegen, A. P., Ward, M. L., Spence, P., England, M. H., 2017. Vertical resolution of baroclinic modes in global ocean models, Ocean Modell.. 113, 50-65, doi:10.1016/j.ocemod.2017.03.012.

von Storch, J.-S., Haak, H., Hertwig, E., Fast, I., 2016. Vertical heat and salt fluxes due to resolved and parameterised meso-scale eddies, Ocean Modell., 108, 1-19, doi:10.1016/j.ocemod.2016.10.001.

---

## Author Comment (AC1) · 5 Jun 2020

Response in supplement

Please also note the supplement to this comment:
https://www.geosci-model-dev-discuss.net/gmd-2019-374/gmd-2019-374-AC1-supplement.pdf

<hr>

2020.

---

## Author Response (AR1)

We thank the reviewers for their insightful comments. The reviewers' comments are repeated below in italics, followed by point-by-point responses to each. We also have improved the figures and text covering sea ice after helpful comments from a collaborator (C. Horvat) who has been added as a co-author. All figures were updated to ensure consistent fonts and notations.

The revised manuscript with tracking of all the changes that have been made is appended at the end of this response.

**Response to Reviewer #1**

*This paper shows the impact of horizontal resolution on ocean-sea-ice model simulations by comparing the outputs from four models at low and high horizontal resolutions. Several biases in the low-resolutions models are improved when the resolution becomes high. The impacts of model resolution on temperature and salinity bias depend on the model. Although the model configuration differences between low and high resolutions differ among the four models, several improvements in high-resolution models compared to low-resolution models seem to be robust. On the one hand, the biases of temperature and salinity fields depend on the model and region. The report in this paper is informative for readers of GMD. I recommend this paper for publication after some minor modifications.*

*Comments*
*The time evolution of the global temperature was examined in Section 3.2. Levitus et al. (2005) showed a warming of the world ocean over 1955-2003 based on the observations. The warmings in the models seem to be consistent with the observation including its decadal variations. As well as the sea surface height, the impact of global warming should be discussed.*

A primary focus of Section 3 is on simulation drift, which tends to be much larger than forced variability in the first few forcing cycles of OMIP simulations for many fields, particularly global ocean temperature (Danabasoglu et al. 2014; Griffies et al. 2014). With only one forcing cycle available from the high-resolution OMIP-2 runs, it can be difficult to distinguish forced variability from model drift. The OMIP simulations are initialized from observed climatologies, not from observed 1958 conditions, and so it is not expected that they should accurately simulate the full-depth evolution of the ocean over the 1958-2018 forcing cycle. In the real world, this evolution reflects the cumulative effect of hundreds or thousands of years of surface forcing. Furthermore, the JRA55-do forcing dataset includes adjustments designed to yield zero net climatological heat/freshwater flux into the ocean for the 1988-2007 time period (Tsujino et al. 2018). Thus, an ocean model that perfectly replicates the observed sea surface state should exhibit zero net ocean heating over that 20-year time period when forced with JRA55-do. This is unrealistic, but it is a reasonable choice in the dataset construction given the large uncertainty in the climatological net surface forcing of the ocean (Tsujino et al. 2018). While it is problematic to assess the realism of long-term ocean warming in OMIP simulations for all the reasons outlined above, the JRA55-do forcing should reflect variations in the global energy imbalance that produce interannual to decadal changes in the rate of ocean warming (Trenberth et al. 2014).

In response to the reviewer, we now include two versions of the original Figure 4 (time evolution of global temperature and salinity) in the paper. The original version (showing anomalies relative to initial conditions – Figure 4 in revised manscript) highlights model drift away from observed ocean climatology. The new version (showing anomalies relative to 2018 – Figure 5 in revised manuscript)

highlights the forced ocean variations of the last few decades of simulation. The comparison with observed heat/salt content change from the WOA18 product in panels c-f of Figure 5 shows that high resolution improves the fidelity and reduces the spread of forced ocean heat content change (particularly 0-2000m heat content) over recent decades. Figures 7-9 in the revised manuscript (time evolution of ocean temperature as a function of depth) are left unchanged because they are intended to illuminate the geographical regions where model drift (i.e., bias growth) is most pronounced, not where forced ocean heat uptake occurs.

**Version 2 of Figure 4, to be included in the manuscript as Figure**

*Figure 9 shows the surface temperature and salinity differences between the simulated fields over 2009-2018 and the initial fields, which discusses about how the simulated field changes from its initial state. Global warming appears to influence much the temperature difference. The model bias compared to the observation should be correctly verified. The simulated surface temperature and salinity are constrained to atmospheric forcing and the restoring to the observation respectively. Because these constrain are not perfect. Therefore, it is worth verifying the surface temperature and salinity biases compared with recent observations such as Argo float in the same period.*

The SST/SSS bias plot (now Fig. 10) has been replotted using concurrent observations (WOA18) spanning 1980-2018 to match Figures 11, 12, and 13 in the revised manuscript. The new plot differs somewhat from the old, but still indicates that model bias is essentially a reflection of model drift away from the initial conditions. This bias generally sets up within a decade or two of initialization (Figures 7-10). The discussion in the paper has been updated accordingly.

*Minor comments*
*There are not a few typos in the texts. The lead-author should be responsible to make readable paper. In Section 2, the model descriptions of four models are shown. It seems that different authors responsible to each model wrote the texts for each configuration, which makes the texts unreadable. The configuration of each model should be written in the same writing style.*

Agreed. The description of the models was revised to be consistent with each other.

*GMD recommend the unit with negative exponent (e.g. m sˆ-1 instead of m/s).*

Corrected throughout the manscript.

*L82: 'AMOC' appears first. 'Atlantic meridional overturning circulation' should be shown.*

Done.

*L98: $\sigma 2$ should be defined.*

Defined.

*L107: $\Delta x$ should be defined such as 'zonal grid spacing'.*

Now defined. HYCOM uses a regular grid on a Mercator projection.

*L145-146: 'm2 s-1' should be modified by using superscript.*

Corrected throughout the manuscript.

*L313: 'section 2b,c' should be 'section 3.2 and 3.3'.*

Corrected.

*Sec.3.4: The surface temperature and salinity averaged over 2009-2018 are compared to the initial state (Fig. 9). On the one hand, the vertical structures averaged over 1980-2018 are compared to the initial state (Fig. 10, 11, 12). Why are the average periods different?*

Good point. In light of the reviewer's earlier comment on section, Figure 9 (now Figure 10) was revised with a comparison to the 1980-2018 average period.

*Sec.3.7: Please plot the EKE time series along the ACC, if possible. Were the EKE in the models were increased by the zonal wind intensification since the 1970s.*

The difficulty with an evaluation of the EKE response to winds is that there is also an initial condition transient in the buoyancy gradients across the ACC due to the model drift discussed above (see Figure 14c illustrating the buoyancy drift). Since the MOC across the ACC is equilibrating for a significant fraction of the whole time of the high-resolution integration, it is difficult to interpret the mechanisms leading to EKE response and it is somewhat beyond the scope of this paper. Thus, we anticipate that this figure would be quite confusing to the readers. In light of this comment and one from Reviewer 2, we have expanded the discussion.

*L580: I cannot find Ollitrault and Colin de Verdière, 2014 in References.*

The reference has been added.

*Fig. 27: Please add the curves of heat transport of IAP-LICOM high resolution in Fig. 27.*

Done.

*'summer' and 'winter' in the captions of Fig. 30 and 31 should be 'winter' and 'summer' respectively.*

Corrected.

*L242, 553, 1170: Renault et al. (2019) should be Renault et al. (2020).*

Corrected.

*L909-918: Please confirm the sequence of references.*

Corrected.

*L963 includes two references.*

Corrected.

*L1204 includes two references.*

Corrected.

*Table.1: Please show the initial conditions.*

The initial conditions were added to Table 1.

*Table.1: 'Relative wind stress' should be 'Absolute wind stress' for FSU-HYCOM'.*

Corrected.

**Response to Reviewer #2**

General Comments
*This is an important paper for the ocean and climate modelling community feeding into wider decisions on the benefits of high resolution ocean components to address climate science questions. The conclusion that the impact of resolution is consistent across model families in terms of circulation but model dependent in terms of temperature and salinity biases is an important one. My main comment is that the high and low resolution versions of the models are not always comparable due to differences in vertical as well as horizontal resolution. It would have been preferable to have changed only the horizontal resolution and associated physics so that the effects of horizontal resolution can be isolated as stated in the abstract. I would suggest that at very least, this issue should be addressed in the model description and discussion in the context of Stewart et al. (Ocean Modelling, 2017) and included as a future recommendation.*

Done. This was an opportunistic study taking advantage of ongoing simulations rather than a dedicated parallel set of simulations. As discussed below, more often than not, the low- and high-resolution were configured independently for different scientific goals and followed different development trajectories, when ideally they should have been designed in tandem.

*I would recommend this paper for publication assuming that this and other specific issues below are addressed.*

Specific comments:
*L21: See discussion above on isolating the effects of horizontal resolution*

This is now discussed in the introduction of Section 2.

*L86: Title would be better as Model Descriptions*

The title has been changed to "Description of the models"

*L93: A summary table of the four models at both resolutions would be useful for the reader*

The four models are summarized in Table 1. The table has been updated to include the climatologies used to initialize the models.

*L117: Why is FSU model 41 layers in low res and 36 layers in high res? (see above)*

The 36-layer high-resolution configuration was at the time our default configuration and we retained it to compare to previous runs performed with other atmospheric forcings. The 41-layer coarse-resolution runs were performed afterward for inclusion in the Tsujino et al. (2020) using the latest vertical grid with all the additional layers in the upper ocean. This is now addressed in the revised manuscript.

*L157: What is the impact in the NCAR model of two additional levels as well as partial step formulation?*

The additional (250 m thick) vertical levels in high resolution POP increase the maximum ocean depth from 5500 m to 6000 m, allowing for a more realistic representation of deep ocean features resolved by the 0.1° grid. The partial bottom cell formulation improves the representation of deep bathymetric slopes. Neither has been tested in the 1° version, and so the impacts are unknown, but are presumed to be small compared to the horizontal grid refinement which includes overall improved representation of small-scale bathymetric features (even without the minor changes to vertical discretization). It should be noted that the bottom depths in the POP model description (section 2.2) were incorrect and should have been 5500m (1-degree) and 6000m (0.1-degree). This has been corrected in the revised manuscript.

*L207: I couldn't see any mention in the description of the lack of salt flux normalization in IAP?*

This has been added.

*L213: What is the impact of changing from 30 to 55 levels in the high resolution?*
*It seems to me that this is more consistent with resolving baroclinic modes at high resolution (Stewart et al., 2017) but 55 levels is probably insufficient to resolve the first baroclinic mode.*

The main goal was to increase the vertical resolution in the deep ocean to improve the simulation of the deep circulation and the AMOC transport. The AMOC transport is better represented at 26.5°N in the high- resolution CAS-LICOM experiment than in the low-resolution experiment (Figure 26). We do agree with the reviewer that 55 levels may be insufficient to resolve the first baroclinic mode, but the 55 levels were dictated by the computing resources at hand. This is now addressed in the revised manuscript.

*L229: Title would be better as Temporal evolution and model drift*

Agreed. The title has been changed accordingly.

*L249: Is there any comment on why the global average KE is so different between NCAR and IAP (which should be fairly comparable in terms of resolution)?*

This is under investigation by the LICOM team, but it most likely due to the numerics, especially the two-step shape preservation advection scheme used in the momentum equations. The Euler-backward finite difference scheme used in the advection scheme is quite diffusive and the LICOM team is exploring ways to decrease the frequency of which the Euler-backward finite difference scheme is used. In addition, CAS-LICOM uses a relatively large horizontal diffusivity. This is now addressed in the revised manuscript.

*L258: Add this discussion of the salt flux normalization to model description and specifically in description of IAP*

Done.

*L275: Please correct 'all high resolution models warm faster over the upper 2000m and global temperature than their lower-resolution partners'. This doesn't appear to be true for FSU.*

Corrected.

*L284: I understand that including this is beyond the scope of this paper, but the authors could recommend and pass to the discussion, that the vertical heat and salt budgets are compared in future as in Griffies et al. (J. Climate, 2015) and Von Storch et al. (Ocean Modelling, 2016)*

Done.

*L335: Is it possible to hypothesise what leads to reductions or increases in the biases and the different model behaviour?*

We can hypothesize that the thermocline bias is related to the representation of vertical eddy heat flux (Griffies et al. 2015), which tends to be stronger and more realistic in high resolution simulations (see revised Figure 4 in response to reviewer #1). The degradation in thermocline bias in POP high resolution could also be due to missing submesoscale physics, which are parameterized in low resolution, but absent in high resolution, versions of that model. This is now addressed in the revised manuscript.

*L417: is the similarity of the high resolution model and RAPID AMOC evolutions statistically significant or can it be shown to link with reduced deep convection? Are the convection patterns different across resolutions? I think without further discussion, this point is rather speculative.*

Most of the discussion highlights in broad terms the similarity between the model multi-decadal variability and the observations as put forward by Hakkinen and Rhines (2004) and Yashayaev (2007) and does not speculate on the RAPID variability. The RAPID time series is only provided for reference and to show consistency among the observations and the models. The text was slightly modified to better convey this point.

*L445: Should figure 14c be labelled as AABW cell?*

We prefer global MOC transport (see Lumpkin and Speer (2007) and Talley (2013) for observed values). Figure 14c (now Figure 15c) will be been relabeled as "global MOC transport at 34°S" and the figure caption modified to reflect that the negative values in global MOC transport are due to the northward-flowing AABW below the southward return flow.

*L502: Since the temperature and salinity fields at depth are still evolving at the end of these experiments, it seems more likely that the ACC changes are related to buoyancy drifts. Can you comment on this?*

There is indeed an initial condition transient in the buoyancy gradients across the ACC (Figure 14c) and the MOC across the ACC is equilibrating for a significant fraction of the whole time of the high-resolution integration. The discussion was extended to reflect this point. See also the response to reviewer #1 regarding the EKE sensitivity to decadal wind stress changes.

*L528: Also comment here about the choice of vertical levels affecting the representation of baroclinic modes (Stewart et al., 2017)*

Done.

*L539: I would argue from figure 16, that there is at least marginal improvement in the representation of the NW corner across the models but it is difficult to see in the multi panel figure*

The representation of the NW corner in these experiments was further discussed in a companion CLIVAR Variations/Exchanges article (Figure 1 of Chassignet et al., 2020). The improvement is very small. This is now referred to in the revised manuscript.

*L555: Please comment on Agulhas eddies in other models*

Done.

*L710: Can you comment on the cause of the increased northward heat transport in the Atlantic by looking at the components? Presumably, it is related to a stronger Gulf Stream transport*

Northward heat transport is usually correlated with higher AMOC transport. In the high-resolution experiments, there is also an overall reduction in subtropical Atlantic upper ocean cold bias (see Figure 23) which is likely to have an impact (Msadek et al., 2013). This has been addressed in the revised manuscript.

*L834: Reference should be Griffies et al. (2015) see below. Include here recommendation on diagnosing vertical heat and salt budgets in future experiments?*

Done.

*L848: I think specific comment is merited here on the aim to maintain traceability across resolution hierarchies in terms of numerics and parameterisations as well as vertical resolution*

Done.

*Technical corrections:*
*L106 (and similar throughout): present as 3 x 10-5*

Corrected.

*L414: extra 'only' in the sentence*

Corrected

*L505: The sentence doesn't make sense as currently written*

Rewritten.

[revised manuscript text omitted]

English (United States), Superscript

| Page 5: [6] Formatted | Eric Chassignet | 6/30/2020 4:16:00 PM |

English (United States)

| Page 5: [7] Formatted | Eric Chassignet | 6/30/2020 4:16:00 PM |

English (United States), Superscript

| Page 5: [8] Formatted | Eric Chassignet | 6/30/2020 4:16:00 PM |

English (United States)

| Page 5: [9] Formatted | Eric Chassignet | 6/30/2020 4:16:00 PM |

English (United States), Superscript

| Page 5: [10] Formatted | Eric Chassignet | 6/30/2020 4:16:00 PM |

English (United States)

| Page 5: [11] Formatted | Eric Chassignet | 6/30/2020 4:16:00 PM |

English (United States), Superscript

| Page 5: [12] Formatted | Eric Chassignet | 6/30/2020 4:16:00 PM |

English (United States)

| Page 5: [13] Formatted | Eric Chassignet | 6/18/2020 5:59:00 PM |

Space After:  0 pt

| Page 8: [14] Deleted | Eric Chassignet | 6/18/2020 6:31:00 PM |
|---|---|---|

The vertical viscosity and diffusion coefficients in the mixed layer have computed by the scheme of Canuto et al. (2001, 2002) with the background values of $2\ 10^{-6}$ m$^2$/s and the upper limit of $2\ 10^{-2}$ m$^2$/s. Recently, a tidal mixing scheme of St. Laurent et al. (2002) has been adopted in LICOM3 by Yu et al. (2017). The Laplacian form with the coefficient of 5400 m$^2$/s was chosen for the horizontal viscosity in the low-resolution version and the biharmonic form with the coefficient of $-2.8\ 10^{10}$ m$^4$/s for the high-resolution experiment. In the low-resolution LICOM3, the isopycnal tracer diffusion scheme of Redi (1982) and the eddy-induced tracer transport scheme of Gent and McWilliams (1990, GM) with the same coefficients are used to parameterize the effects of mesoscale eddies on the large-scale circulation. Two tapering factors of Large et al. (1997) and a buoyancy frequency ($N^2$) related thickness diffusivity of Ferreira et al. (2005) are also employed. However, for the high resolution, GM

| Page 8: [15] Deleted | Eric Chassignet | 6/18/2020 6:33:00 PM |
|---|---|---|

Besides, the chlorophyll-a dependent solar penetration of Ohlmann (2003) introduced by Lin et al. (2007) was also implemented in both simulations.

| Page 8: [16] Deleted | Eric Chassignet | 6/18/2020 6:33:00 PM |
|---|---|---|

are employed for both low- and high-resolution experiments following the OMIP protocol. However, onl

| Page 13: [17] Deleted | Eric Chassignet | 6/22/2020 3:07:00 PM |
|---|---|---|

| Page 13: [17] Deleted | Eric Chassignet | 6/22/2020 3:07:00 PM |
|---|---|---|

| Page 13: [17] Deleted | Eric Chassignet | 6/22/2020 3:07:00 PM |
|---|---|---|

| Page 13: [18] Deleted | Eric Chassignet | 6/22/2020 3:07:00 PM |
|---|---|---|

| Page 13: [18] Deleted | Eric Chassignet | 6/22/2020 3:07:00 PM |
|---|---|---|

| Page 13: [18] Deleted | Eric Chassignet | 6/22/2020 3:07:00 PM |
|---|---|---|

| Page 19: [19] Deleted | Eric Chassignet | 6/17/2020 11:48:00 AM |
|---|---|---|

Sea ice is usually quantified and monitored by satellite in terms of sea ice extent, defined as the area with 15% or higher sea-ice concentration. Figures 13a-b display the modeled (annual mean) northern and southern hemisphere sea-ice extent for all simulations and are compared to the latest observations from the National Snow and Ice Data Center (NSIDC, Fetterer et al., 2017). Despite a wide range of time mean sea-ice extent, all simulations represent well the observed variability of the sea-ice extent from 1979 to 2018. In the northern hemisphere, the sea-ice extent shows a clear decline since the beginning of the monitored period, whereas, in the southern hemisphere, the sea ice extent remains more or less stable until the last few years of the integration with a small upward trend. Only the lowresolution version of FSU-HYCOM shows an inconsistent negative trend in the southern hemisphere sea-ice extent. It is known that the observed sea-ice extent is highly correlated with the near-surface air temperature (e.g., Olonscheck et al., 2019). Thus, the consistency of the temporal variability between different simulations and observations suggests that the air-temperature in the JRA55-do is quite realistic. It is also notable that in the northern hemisphere, all of the high-resolution models show less bias in sea-ice extent than their low-resolution counterparts. In the southern hemisphere, all of the high-resolution models show less bias in extent than their low-resolution counterparts, with the exception of IAP-LICOM.

| Page 22: [20] Formatted | Eric Chassignet | 6/30/2020 4:16:00 PM |
|---|---|---|

English (United States)

| Page 22: [20] Formatted | Eric Chassignet | 6/30/2020 4:16:00 PM |
|---|---|---|

English (United States)

| Page 22: [20] Formatted | Eric Chassignet | 6/30/2020 4:16:00 PM |
|---|---|---|

English (United States)

| Page 39: [21] Deleted | Eric Chassignet | 6/22/2020 3:40:00 PM |
|---|---|---|

| Page 39: [21] Deleted | Eric Chassignet | 6/22/2020 3:40:00 PM |
|---|---|---|

| Page 39: [21] Deleted | Eric Chassignet | 6/22/2020 3:40:00 PM |
|---|---|---|

| Page 39: [21] Deleted | Eric Chassignet | 6/22/2020 3:40:00 PM |
|---|---|---|

| Page 39: [21] Deleted | Eric Chassignet | 6/22/2020 3:40:00 PM |
|---|---|---|

| Page 39: [21] Deleted | Eric Chassignet | 6/22/2020 3:40:00 PM |
|---|---|---|

| Page 39: [22] Deleted | Eric Chassignet | 6/17/2020 4:10:00 PM |
|---|---|---|

-

| Page 39: [22] Deleted | Eric Chassignet | 6/17/2020 4:10:00 PM |
|---|---|---|

-

| Page 39: [23] Formatted | Eric Chassignet | 6/30/2020 4:16:00 PM |
|---|---|---|

English (United States)

| Page 39: [23] Formatted | Eric Chassignet | 6/30/2020 4:16:00 PM |
|---|---|---|

English (United States)

**Page 39: [23] Formatted** Eric Chassignet 6/30/2020 4:16:00 PM

English (United States)

**Page 39: [23] Formatted** Eric Chassignet 6/30/2020 4:16:00 PM

English (United States)

**Page 39: [23] Formatted** Eric Chassignet 6/30/2020 4:16:00 PM

English (United States)

**Page 39: [23] Formatted** Eric Chassignet 6/30/2020 4:16:00 PM

English (United States)

**Page 39: [23] Formatted** Eric Chassignet 6/30/2020 4:16:00 PM

English (United States)

**Page 39: [23] Formatted** Eric Chassignet 6/30/2020 4:16:00 PM

English (United States)

**Page 39: [23] Formatted** Eric Chassignet 6/30/2020 4:16:00 PM

English (United States)

**Page 39: [23] Formatted** Eric Chassignet 6/30/2020 4:16:00 PM

English (United States)

**Page 39: [23] Formatted** Eric Chassignet 6/30/2020 4:16:00 PM

English (United States)

**Page 39: [23] Formatted** Eric Chassignet 6/30/2020 4:16:00 PM

English (United States)

**Page 39: [23] Formatted** Eric Chassignet 6/30/2020 4:16:00 PM

English (United States)

**Page 39: [23] Formatted** Eric Chassignet 6/30/2020 4:16:00 PM

English (United States)

**Page 39: [23] Formatted** Eric Chassignet 6/30/2020 4:16:00 PM

English (United States)

**Page 39: [23] Formatted** Eric Chassignet 6/30/2020 4:16:00 PM

English (United States)

**Page 39: [23] Formatted** Eric Chassignet 6/30/2020 4:16:00 PM

English (United States)

| Page 39: [23] Formatted | Eric Chassignet | 6/30/2020 4:16:00 PM |
| --- | --- | --- |

English (United States)

| Page 39: [24] Formatted | Eric Chassignet | 6/30/2020 4:16:00 PM |
| --- | --- | --- |

English (United States)

| Page 39: [24] Formatted | Eric Chassignet | 6/30/2020 4:16:00 PM |

English (United States)

| Page 39: [24] Formatted | Eric Chassignet | 6/30/2020 4:16:00 PM |

English (United States)

| Page 39: [24] Formatted | Eric Chassignet | 6/30/2020 4:16:00 PM |

English (United States)

| Page 39: [24] Formatted | Eric Chassignet | 6/30/2020 4:16:00 PM |

English (United States)

| Page 39: [24] Formatted | Eric Chassignet | 6/30/2020 4:16:00 PM |

English (United States)

[revised manuscript text omitted]

English (United States)

| Page 60: [30] Formatted | Eric Chassignet | 6/30/2020 4:16:00 PM |
|---|---|---|

English (United States)

| Page 60: [30] Formatted | Eric Chassignet | 6/30/2020 4:16:00 PM |
|---|---|---|

English (United States)

| Page 60: [30] Formatted | Eric Chassignet | 6/30/2020 4:16:00 PM |
|---|---|---|

English (United States)

| Page 60: [30] Formatted | Eric Chassignet | 6/30/2020 4:16:00 PM |
|---|---|---|

English (United States)

| Page 60: [30] Formatted | Eric Chassignet | 6/30/2020 4:16:00 PM |
|---|---|---|

English (United States)

| Page 60: [31] Formatted | Eric Chassignet | 6/30/2020 4:16:00 PM |
|---|---|---|

English (United States)

| Page 60: [31] Formatted | Eric Chassignet | 6/30/2020 4:16:00 PM |
|---|---|---|

English (United States)

| Page 60: [31] Formatted | Eric Chassignet | 6/30/2020 4:16:00 PM |
|---|---|---|

English (United States)

| Page 60: [32] Formatted | Eric Chassignet | 6/30/2020 4:16:00 PM |
|---|---|---|

English (United States)

| Page 60: [32] Formatted | Eric Chassignet | 6/30/2020 4:16:00 PM |
|---|---|---|

English (United States)

| Page 60: [32] Formatted | Eric Chassignet | 6/30/2020 4:16:00 PM |
|---|---|---|

English (United States)

| Page 60: [33] Deleted | Eric Chassignet | 6/18/2020 2:56:00 PM |
|---|---|---|

Relative

| Page 60: [33] Deleted | Eric Chassignet | 6/18/2020 2:56:00 PM |
|---|---|---|

Relative

| Page 60: [34] Formatted | Eric Chassignet | 6/30/2020 4:16:00 PM |
|---|---|---|

English (United States)

| Page 60: [34] Formatted | Eric Chassignet | 6/30/2020 4:16:00 PM |
|---|---|---|

English (United States)

| Page 60: [34] Formatted | Eric Chassignet | 6/30/2020 4:16:00 PM |
|---|---|---|

English (United States)

| Page 60: [34] Formatted | Eric Chassignet | 6/30/2020 4:16:00 PM |
|---|---|---|

English (United States)

| Page 60: [34] Formatted | Eric Chassignet | 6/30/2020 4:16:00 PM |
|---|---|---|

English (United States)

| Page 60: [34] Formatted | Eric Chassignet | 6/30/2020 4:16:00 PM |
|---|---|---|

English (United States)

| Page 60: [34] Formatted | Eric Chassignet | 6/30/2020 4:16:00 PM |
|---|---|---|

English (United States)

| Page 60: [34] Formatted | Eric Chassignet | 6/30/2020 4:16:00 PM |
|---|---|---|

English (United States)

| Page 60: [35] Formatted | Eric Chassignet | 6/30/2020 4:16:00 PM |
|---|---|---|

English (United States)

| Page 60: [35] Formatted | Eric Chassignet | 6/30/2020 4:16:00 PM |
|---|---|---|

English (United States)

| Page 60: [35] Formatted | Eric Chassignet | 6/30/2020 4:16:00 PM |
|---|---|---|

English (United States)

| Page 60: [36] Formatted | Eric Chassignet | 6/30/2020 4:16:00 PM |
|---|---|---|

English (United States)

| Page 60: [36] Formatted | Eric Chassignet | 6/30/2020 4:16:00 PM |
|---|---|---|

English (United States)

| Page 60: [36] Formatted | Eric Chassignet | 6/30/2020 4:16:00 PM |
|---|---|---|

English (United States)

| Page 60: [37] Deleted | Eric Chassignet | 6/18/2020 2:56:00 PM |
|---|---|---|

Relative

| Page 60: [37] Deleted | Eric Chassignet | 6/18/2020 2:56:00 PM |
|---|---|---|

Relative

---

## Author Response (AR3)

**Comments to the Author:**
At the early stage of a MIP it is quite common for parts of the protocol to not be defined, or for ad-hoc ensembles to be formed from existing runs or already published runs. This is actually an important thing as it enables experimentation with what protocols work well, which could then be more clearly defined (or not!) in further comparisons. Your MIP is a mixture of all these things. This was explained in your response to me, but you need to add it to the manuscript so that readers not involved in the MIP can also understand the context of the paper.

This is the response I am referring to:
"Ocean-only models are used either to develop the ocean component of a specific climate model or to address specific ocean processes. They are configured using best practices, but each modeling group was empowered to choose what they think is best and that includes initial conditions. The high-resolution experiments are computationally expensive and, when the call for comparison was made, each group leveraged known and proven configurations to perform the requested experiments. Furthermore, some groups had already completed the JRA55-do simulations at high-resolution before this intercomparison was conceived. Given the large computational resources involved, rerunning those experiments to conform to a standard protocol was not an option. "

The points I would like to see emphasised in the paper at the place where the ensemble is introduced are: that some runs were pre-existing; that modellers were free to choose own boundary conditions; and that the runs are presently very computationally expensive.

**Response:**

The above points were incorporated in the revised paper as recommended (see below). Thank you again for handling this manuscript.

[revised manuscript text omitted]